# Crystal Structure Prediction
# by Joint Equivariant Diffusion

**Rui Jiao**[1,2] **Wenbing Huang**[3,4*] **Peijia Lin**[5] **Jiaqi Han**[6] **Pin Chen**[5] **Yutong Lu**[5] **Yang Liu**[1,2*]
[1]Dept. of Comp. Sci. & Tech., Institute for AI, Tsinghua University
[2]Institute for AIR, Tsinghua University
[3]Gaoling School of Artificial Intelligence, Renmin University of China
[4] Beijing Key Laboratory of Big Data Management and Analysis Methods, Beijing, China
[5] National Supercomputer Center in Guangzhou,
School of Computer Science and Engineering, Sun Yat-sen University
[6] Stanford University

## Abstract

Crystal Structure Prediction (CSP) is crucial in various scientific disciplines. While CSP can be addressed by employing currently-prevailing generative models (*e.g.* diffusion models), this task encounters unique challenges owing to the symmetric geometry of crystal structures—the invariance of translation, rotation, and periodicity. To incorporate the above symmetries, this paper proposes DiffCSP, a novel diffusion model to learn the structure distribution from stable crystals. To be specific, DiffCSP jointly generates the lattice and atom coordinates for each crystal by employing a periodic-E(3)-equivariant denoising model, to better model the crystal geometry. Notably, different from related equivariant generative approaches, DiffCSP leverages fractional coordinates other than Cartesian coordinates to represent crystals, remarkably promoting the diffusion and the generation process of atom positions. Extensive experiments verify that our DiffCSP significantly outperforms existing CSP methods, with a much lower computation cost in contrast to DFT-based methods. Moreover, the superiority of DiffCSP is also observed when it is extended for ab initio crystal generation. Code is available at `https://github.com/jiaor17/DiffCSP`.

## 1 Introduction

Crystal Structure Prediction (CSP), which returns the stable 3D structure of a compound based solely on its composition, has been a goal in physical sciences since the 1950s [1]. As crystals are the foundation of various materials, estimating their structures in 3D space determines the physical and chemical properties that greatly influence the application to various academic and industrial sciences, such as the design of drugs, batteries, and catalysts [2]. Conventional methods towards CSP mostly apply the Density Functional Theory (DFT) [3] to compute the energy at each iteration, guided by optimization algorithms (such as random search [4], Bayesian optimization [5], etc.) to iteratively search for the stable state corresponding to the local minima of the energy surface [6].

The DFT-based approaches are computationally-intensive. Recent attention has been paid to deep generative models, which directly learn the distribution from the training data consisting of stable structures [7, 8]. More recently, diffusion models, a special kind of deep generative models are employed for crystal generation [9], encouraged by their better physical interpretability and enhanced performance than other generative models. Intuitively, by conducting diffusion on stable structures,

---

*Wenbing Huang and Yang Liu are corresponding authors.

the denoising process in diffusion models acts like a force field that drives the atom coordinates towards the energy local minimum and thus is able to increase stability. Indeed, the success of diffusion models is observed in broad scientific domains, including molecular conformation generation [10], protein structure prediction [11] and protein docking [12].

However, designing diffusion models for CSP is challenging. From the perspective of physics, any E(3) transformation, including translation, rotation, and reflection, of the crystal coordinates does not change the physical law and thus keeps the crystal distribution invariant. In other words, the generation process we design should yield E(3) invariant samples. Moreover, in contrast to other types of structures such as small molecules [13] and proteins [14], CSP exhibits unique challenges, mainly incurred by the periodicity of the atom arrangement in crystals. Figure 1 displays a crystal where the atoms in a unit cell are repeated infinitely in space. We identify such unique symmetry, jointly consisting of E(3) invariance and periodicity, as *periodic E(3) invariance* in this paper. Generating such type of structures requires not only modeling the distribution of the atom coordinates within every cell, but also inferring how their bases (*a.k.a.* lattice vectors) are placed in 3D space. Interestingly, as we will show in § 4.1, such view offers a natural disentanglement for fulfilling the periodic E(3) invariance by separately enforcing constraints on fractional coordinates and lattice vectors, which permits a feasible implementation to encode the crystal symmetry.

In this work, we introduce DiffCSP, an equivariant diffusion method to address CSP. Considering the specificity of the crystal geometry, our DiffCSP jointly generates the lattice vectors and the fractional coordinates of all atoms, by employing a proposed denoising model that is theoretically proved to generate periodic-E(3)-invariant samples. A preferable characteristic of DiffCSP is that it leverages the fractional coordinate system (defined in § 3) other than the Cartesian system used in previous methods to represent crystals [9, 15], which encodes periodicity intrinsically. In particular, the fractional representation not only allows us to consider Wrapped Normal (WN) distribution [16] to better model the periodicity, but also facilitates the design of the denoising model via the Fourier transformation, compared to the traditional multi-graph encoder in crystal modeling [15].

CDVAE [9] is closely related with our paper. It adopts an equivariant Variational Auto-Encoder (VAE) based framework to learn the data distribution and then generates crystals in a score-matching-based diffusion process. However, CDVAE focuses mainly on ab initio crystal generation where the composition is also randomly sampled, which is distinct from the CSP task in this paper. Moreover, while CDVAE first predicts the lattice and then updates the coordinates with the fixed lattice, we jointly update the lattice and coordinates to better model the crystal geometry. Besides, CDVAE represents crystals by Cartesian coordinates upon multi-graph modeling, whereas our DiffCSP applies fractional coordinates without multi-graph modeling as mentioned above.

To sum up, our contributions are as follows:

- To the best of our knowledge, we are the first to apply equivariant diffusion-based methods to address CSP. The proposed DiffCSP is more insightful than current learning-based approaches as the periodic E(3) invariance has been delicately considered.

- DiffCSP conducts joint diffusion on lattices and fractional coordinates, which is capable of capturing the crystal geometry as a whole. Besides, the usage of fractional coordinates in place of Cartesian coordinates used in previous methods (*e.g.* CDVAE [9]) remarkably promotes the diffusion and the generation process of atom positions.

- We verify the efficacy of DiffCSP on the CSP task against learning-based and DFT-based methods, and sufficiently ablate each proposed component in DiffCSP. We further extend DiffCSP into ab initio generation and show its effectiveness against related methods.

## 2 Related Works

**Crystal Structure Prediction** Traditional computation methods [4, 5, 17, 18] combine DFT [3] with optimization algorithms to search for local minima in the potential energy surface. However, DFT is computationally intensive, making it dilemmatic to balance efficiency and accuracy. With the improvement of crystal databases, machine-learning methods are applied as alternative energy predictors to DFT followed by optimization steps [19, 20, 21]. Apart from the predict-optimize paradigm, another line of approaches directly learns stable structures from data by deep generative models, which represents crystals by 3D voxels [7, 22, 23], distance matrices [8, 24, 25] or 3D

coordinates [26, 27, 28]. Unfortunately, these methods are unaware of the full symmetries of the crystal structure. CDVAE [9] has taken the required symmetries into account. However, as mentioned above, the initial version of CDVAE is for different task and utilizes different generation process.

**Equivariant Graph Neural Networks** Geometrically equivariant Graph Neural Networks (GNNs) that ensure E(3) symmetry are powerful tools to represent physical objects [29, 30, 31, 32, 33], and have showcased the superiority in modeling 3D structures [34, 35]. To further model the periodic materials, Xie and Grossman [15] propose the multi-graph edge construction to capture the periodicity by connecting the edges between adjacent lattices. Yan et al. [36] further introduce periodic pattern encoding into a Transformer-based backbone. In this work, we achieve the periodic invariance by introducing the Fourier transform on fractional coordinates.

**Diffusion Generative Models** Motivated by the non-equilibrium thermodynamics [37], diffusion models connect the data distribution with the prior distribution via forward and backward Markov chains [38], and have made remarkable progress in the field of image generation [39, 40]. Equipped with equivariant GNNs, diffusion models are capable of generating samples from the invariant distribution, which is desirable in conformation generation [10, 13], ab initio molecule design [41], protein generation [42], and so on. Recent works extend the diffusion models onto Riemann manifolds [43, 44], and enable the generation of periodic features like torsion angles [12, 16].

# 3 Preliminaries

**Representation of crystal structures** A 3D crystal can be represented as the infinite periodic arrangement of atoms in 3D space, and the smallest repeating unit is called a *unit cell*, as shown in Figure 1. A unit cell can be defined by a triplet $\mathcal{M} = (\boldsymbol{A}, \boldsymbol{X}, \boldsymbol{L})$, where $\boldsymbol{A} = [\boldsymbol{a}_1, \boldsymbol{a}_2, ..., \boldsymbol{a}_N] \in \mathbb{R}^{h \times N}$ denotes the list of the one-hot representations of atom types, $\boldsymbol{X} = [\boldsymbol{x}_1, \boldsymbol{x}_2, ..., \boldsymbol{x}_N] \in \mathbb{R}^{3 \times N}$ consists of Cartesian coordinates of the atoms, and $\boldsymbol{L} = [\boldsymbol{l}_1, \boldsymbol{l}_2, \boldsymbol{l}_3] \in \mathbb{R}^{3 \times 3}$ represents the lattice matrix containing three basic vectors to describe the periodicity of the crystal. The infinite periodic crystal structure is represented by

$$\{(\boldsymbol{a}_i', \boldsymbol{x}_i') | \boldsymbol{a}_i' = \boldsymbol{a}_i, \boldsymbol{x}_i' = \boldsymbol{x}_i + \boldsymbol{L}\boldsymbol{k}, \ \forall \boldsymbol{k} \in \mathbb{Z}^{3 \times 1}\}, \tag{1}$$

where the $j$-th element of the integral vector $\boldsymbol{k}$ denotes the integral 3D translation in units of $\boldsymbol{l}_j$.

**Fractional coordinate system** The Cartesian coordinate system $\boldsymbol{X}$ leverages three standard orthogonal bases as the coordinate axes. In crystallography, the fractional coordinate system is usually applied to reflect the periodicity of the crystal structure [26, 27, 28, 45], which utilizes the lattices $(\boldsymbol{l}_1, \boldsymbol{l}_2, \boldsymbol{l}_3)$ as the bases. In this way, a point represented by the fractional coordinate vector $\boldsymbol{f} = [f_1, f_2, f_3]^\top \in [0, 1)^3$ corresponds to the Cartesian vector $\boldsymbol{x} = \sum_{i=1}^3 f_i \boldsymbol{l}_i$. This paper employs the fractional coordinate system, and denotes the crystal by $\mathcal{M} = (\boldsymbol{A}, \boldsymbol{F}, \boldsymbol{L})$, where the fractional coordinates of all atoms in a cell compose the matrix $\boldsymbol{F} \in [0, 1)^{3 \times N}$.

**Task definition** CSP predicts for each unit cell the lattice matrix $\boldsymbol{L}$ and the fractional matrix $\boldsymbol{F}$ given its chemical composition $\boldsymbol{A}$, namely, learning the conditional distribution $p(\boldsymbol{L}, \boldsymbol{F} \mid \boldsymbol{A})$.

# 4 The Proposed Method: DiffCSP

This section first presents the symmetries of the crystal geometry, and then introduces the joint equivaraint diffusion process on $\boldsymbol{L}$ and $\boldsymbol{F}$, followed by the architecture of the denoising function.

## 4.1 Symmetries of Crystal Structure Distribution

While various generative models can be utilized to address CSP, this task encounters particular challenges, including constraints arising from symmetries of crystal structure distribution. Here, we consider the three types of symmetries in the distribution of $p(\boldsymbol{L}, \boldsymbol{F} \mid \boldsymbol{A})$: permutation invariance, $O(3)$ invariance, and periodic translation invariance. Their detailed definitions are provided as follows.

**Definition 1** (Permutation Invariance)**.** *For any permutation $\boldsymbol{P} \in \mathrm{S}_N$, $p(\boldsymbol{L}, \boldsymbol{F} \mid \boldsymbol{A}) = p(\boldsymbol{L}, \boldsymbol{F}\boldsymbol{P} \mid \boldsymbol{A}\boldsymbol{P})$, i.e., changing the order of atoms will not change the distribution.*

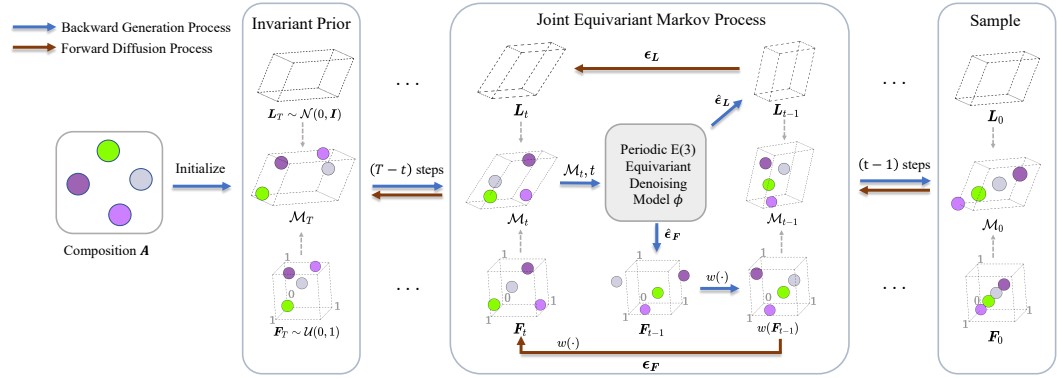

Figure 2: Overview of DiffCSP. Given the composition $\boldsymbol{A}$, we denote the crystal, its lattice and fractional coordinate matrix at time $t$ as $\mathcal{M}_t$, $\boldsymbol{L}_t$ and $\boldsymbol{F}_t$, respectively. The terms $\boldsymbol{\epsilon_L}$ and $\boldsymbol{\epsilon_F}$ are Gaussian noises, $\hat{\boldsymbol{\epsilon}}_{\boldsymbol{L}}$ and $\hat{\boldsymbol{\epsilon}}_{\boldsymbol{F}}$ are predicted by the denoising model $\phi$.

**Definition 2** (O(3) Invariance). *For any orthogonal transformation $\boldsymbol{Q} \in \mathbb{R}^{3 \times 3}$ satisfying $\boldsymbol{Q}^\top \boldsymbol{Q} = \boldsymbol{I}$, $p(\boldsymbol{Q L}, \boldsymbol{F} \mid \boldsymbol{A}) = p(\boldsymbol{L}, \boldsymbol{F} \mid \boldsymbol{A})$, namely, any rotation/reflection of $\boldsymbol{L}$ keeps the distribution unchanged.*

**Definition 3** (Periodic Translation Invariance). *For any translation $\boldsymbol{t} \in \mathbb{R}^{3 \times 1}$, $p(\boldsymbol{L}, w(\boldsymbol{F} + \boldsymbol{t}\mathbf{1}^\top) \mid \boldsymbol{A}) = p(\boldsymbol{L}, \boldsymbol{F} \mid \boldsymbol{A})$, where the function $w(\boldsymbol{F}) = \boldsymbol{F} - \lfloor \boldsymbol{F} \rfloor \in [0, 1)^{3 \times N}$ returns the fractional part of each element in $\boldsymbol{F}$, and $\mathbf{1} \in \mathbb{R}^{3 \times 1}$ is a vector with all elements set to one. It explains that any periodic translation of $\boldsymbol{F}$ will not change the distribution[2].*

The permutation invariance is tractably encapsulated by using GNNs as the backbone for generation [47]. We mainly focus on the other two kinds of invariance (see Figure 1), since GNNs are our default choices. For simplicity, we compactly term the $O(3)$ invariance and periodic translation invariance as *periodic E(3) invariance* henceforth. Previous approaches (*e.g.* [9, 36]) adopt Cartesian coordinates $\boldsymbol{X}$ other than fractional coordinates $\boldsymbol{F}$, hence their derived forms of the symmetry are different. Particularly, in Definition 2, the orthogonal transformation additionally acts on $\boldsymbol{X}$; in Definition 3, the periodic translation $w(\boldsymbol{F} + \boldsymbol{t}\mathbf{1}^\top)$ becomes the translation along the lattice bases $\boldsymbol{X} + \boldsymbol{L}\boldsymbol{t}\mathbf{1}^\top$; besides, $\boldsymbol{X}$ should also maintain E(3) translation invariance, that is $p(\boldsymbol{L}, \boldsymbol{X} + \boldsymbol{t}\mathbf{1}^\top | \boldsymbol{A}) = p(\boldsymbol{L}, \boldsymbol{X} | \boldsymbol{A})$. With the help of the fractional system, the periodic E(3) invariance

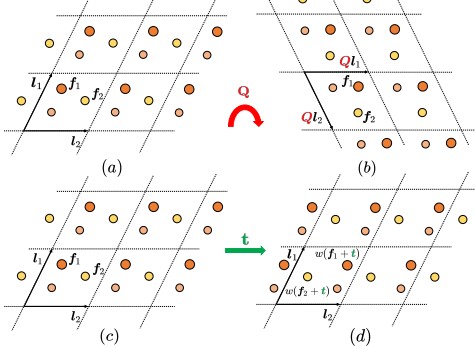

Figure 1: (a)→(b): The orthogonal transformation of the lattice vectors. (c)→(d): The periodic translation of the fractional coordinates. Both cases do not change the structure.

is made tractable by fulfilling O(3) invariance *w.r.t.* the orthogonal transformations on $\boldsymbol{L}$ and periodic translation invariance *w.r.t.* the periodic translations on $\boldsymbol{F}$, respectively. In this way, such approach, as detailed in the next section, facilitates the application of diffusion methods to the CSP task.

## 4.2 Joint Equivariant Diffusion

Our method DiffCSP addresses CSP by simultaneously diffusing the lattice $\boldsymbol{L}$ and the fractional coordinate matrix $\boldsymbol{F}$. Given the atom composition $\boldsymbol{A}$, $\mathcal{M}_t$ denotes the intermediate state of $\boldsymbol{L}$ and $\boldsymbol{F}$ at time step $t$ ($0 \leq t \leq T$). DiffCSP defines two Markov processes: the forward diffusion process gradually adds noise to $\mathcal{M}_0$, and the backward generation process iteratively samples from the prior distribution $\mathcal{M}_T$ to recover the origin data $\mathcal{M}_0$.

---

[2]Previous works (*e.g.* [36]) further discuss the scaling invariance of a unit cell formed by periodic boundaries, allowing $\boldsymbol{L} \to \alpha\boldsymbol{L}, \forall \alpha \in \mathbb{N}_+^3$. In this paper, the scaling invariance is unnecessary since we apply the Niggli reduction [46] on the primitive cell as a canonical scale representation of the lattice vectors where we fix $\alpha = (1, 1, 1)^\top$. Additionally, periodic translation invariance in our paper is equivalent to the invariance of shifting periodic boundaries in [36]. We provide more discussions in Appendix A.4.

Joining the statements in § 4.1, the recovered distribution from $\mathcal{M}_T$ should meet periodic E(3) invariance. Such requirement is satisfied if the prior distribution $p(\mathcal{M}_T)$ is invariant and the Markov transition $p(\mathcal{M}_{t-1} \mid \mathcal{M}_t)$ is equivariant, according to the diffusion-based generation literature [10]. Here, an equivariant transition is specified as $p(g \cdot \mathcal{M}_{t-1} \mid g \cdot \mathcal{M}_t) = p(\mathcal{M}_{t-1} \mid \mathcal{M}_t)$ where $g \cdot \mathcal{M}$ refers to any orthogonal/translational transformation $g$ acts on $\mathcal{M}$ in the way presented in Definitions 2-3. We separately explain the derivation details of $\boldsymbol{L}$ and $\boldsymbol{F}$ below. The detailed flowcharts are summarized in Algorithms 1 and 2 in Appendix B.3.

**Diffusion on $\boldsymbol{L}$** Given that $\boldsymbol{L}$ is a continuous variable, we exploit Denoising Diffusion Probabilistic Model (DDPM) [38] to accomplish the generation. We define the forward process that progressively diffuses $\boldsymbol{L}_0$ towards the Normal prior $p(\boldsymbol{L}_T) = \mathcal{N}(0, \boldsymbol{I})$ by $q(\boldsymbol{L}_t|\boldsymbol{L}_{t-1})$ which can be devised as the probability conditional on the initial distribution:

$$q(\boldsymbol{L}_t|\boldsymbol{L}_0) = \mathcal{N}\Big(\boldsymbol{L}_t|\sqrt{\bar{\alpha}_t}\boldsymbol{L}_0, (1 - \bar{\alpha}_t)\boldsymbol{I}\Big), \tag{2}$$

where $\beta_t \in (0, 1)$ controls the variance, and $\bar{\alpha}_t = \prod_{s=1}^{t} \alpha_t = \prod_{s=1}^{t}(1 - \beta_t)$ is valued in accordance to the cosine scheduler [48].

The backward generation process is given by:

$$p(\boldsymbol{L}_{t-1}|\mathcal{M}_t) = \mathcal{N}(\boldsymbol{L}_{t-1}|\mu(\mathcal{M}_t), \sigma^2(\mathcal{M}_t)\boldsymbol{I}), \tag{3}$$

where $\mu(\mathcal{M}_t) = \frac{1}{\sqrt{\alpha_t}}\Big(\boldsymbol{L}_t - \frac{\beta_t}{\sqrt{1-\bar{\alpha}_t}}\hat{\boldsymbol{\epsilon}}_{\boldsymbol{L}}(\mathcal{M}_t, t)\Big), \sigma^2(\mathcal{M}_t) = \beta_t \frac{1-\bar{\alpha}_{t-1}}{1-\bar{\alpha}_t}$. The denoising term $\hat{\boldsymbol{\epsilon}}_{\boldsymbol{L}}(\mathcal{M}_t, t) \in \mathbb{R}^{3\times3}$ is predicted by the model $\phi(\boldsymbol{L}_t, \boldsymbol{F}_t, \boldsymbol{A}, t)$.

As the prior distribution $p(\boldsymbol{L}_T) = \mathcal{N}(0, \boldsymbol{I})$ is already O(3)-invariant, we require the generation process in Eq. (3) to be O(3)-equivariant, which is formally stated below.

**Proposition 1.** *The marginal distribution $p(\boldsymbol{L}_0)$ by Eq. (3) is O(3)-invariant if $\hat{\boldsymbol{\epsilon}}_{\boldsymbol{L}}(\mathcal{M}_t, t)$ is O(3)-equivariant, namely $\hat{\boldsymbol{\epsilon}}_{\boldsymbol{L}}(\boldsymbol{Q}\boldsymbol{L}_t, \boldsymbol{F}_t, \boldsymbol{A}, t) = \boldsymbol{Q}\hat{\boldsymbol{\epsilon}}_{\boldsymbol{L}}(\boldsymbol{L}_t, \boldsymbol{F}_t, \boldsymbol{A}, t), \forall \boldsymbol{Q}^\top \boldsymbol{Q} = \boldsymbol{I}$.*

To train the denoising model $\phi$, we first sample $\boldsymbol{\epsilon}_{\boldsymbol{L}} \sim \mathcal{N}(0, \boldsymbol{I})$ and reparameterize $\boldsymbol{L}_t = \sqrt{\bar{\alpha}_t}\boldsymbol{L}_0 + \sqrt{1 - \bar{\alpha}_t}\boldsymbol{\epsilon}_{\boldsymbol{L}}$ based on Eq. (2). The training objective is defined as the $\ell_2$ loss between $\boldsymbol{\epsilon}_{\boldsymbol{L}}$ and $\hat{\boldsymbol{\epsilon}}_{\boldsymbol{L}}$:

$$\mathcal{L}_{\boldsymbol{L}} = \mathbb{E}_{\boldsymbol{\epsilon}_{\boldsymbol{L}} \sim \mathcal{N}(0, \boldsymbol{I}), t \sim \mathcal{U}(1, T)}[\|\boldsymbol{\epsilon}_{\boldsymbol{L}} - \hat{\boldsymbol{\epsilon}}_{\boldsymbol{L}}(\mathcal{M}_t, t)\|_2^2]. \tag{4}$$

**Diffusion on $\boldsymbol{F}$** The domain of fractional coordinates $[0, 1)^{3\times N}$ forms a quotient space $\mathbb{R}^{3\times N}/\mathbb{Z}^{3\times N}$ induced by the crystal periodicity. It is not suitable to apply the above DDPM fashion to generate $\boldsymbol{F}$, as the normal distribution used in DDPM is unable to model the cyclical and bounded domain of $\boldsymbol{F}$. Instead, we leverage Score-Matching (SM) based framework [49, 50] along with Wrapped Normal (WN) distribution [43] to fit the specificity here. Note that WN distribution has been explored in generative models, such as molecular conformation generation [16].

During the forward process, we first sample each column of $\boldsymbol{\epsilon}_{\boldsymbol{F}} \in \mathbb{R}^{3\times N}$ from $\mathcal{N}(0, \boldsymbol{I})$, and then acquire $\boldsymbol{F}_t = w(\boldsymbol{F}_0 + \sigma_t\boldsymbol{\epsilon}_{\boldsymbol{F}})$ where the truncation function $w(\cdot)$ is already defined in Definition 3. This truncated sampling implies the WN transition:

$$q(\boldsymbol{F}_t|\boldsymbol{F}_0) \propto \sum_{\boldsymbol{Z} \in \mathbb{Z}^{3\times N}} \exp\Big(-\frac{\|\boldsymbol{F}_t - \boldsymbol{F}_0 + \boldsymbol{Z}\|_F^2}{2\sigma_t^2}\Big). \tag{5}$$

Basically, this process ensures the probability distribution over $[z, z + 1)^{3\times N}$ for any integer $z$ to be the same to keep the crystal periodicity. Here, the noise scale $\sigma_t$ obeys the exponential scheduler: $\sigma_0 = 0$ and $\sigma_t = \sigma_1(\frac{\sigma_T}{\sigma_1})^{\frac{t-1}{T-1}}$, if $t > 0$. Desirably, $q(\boldsymbol{F}_t|\boldsymbol{F}_0)$ is periodic translation equivariant, and approaches a uniform distribution $\mathcal{U}(0, 1)$ if $\sigma_T$ is sufficiently large.

For the backward process, we first initialize $\boldsymbol{F}_T$ from the uniform distribution $\mathcal{U}(0, 1)$, which is periodic translation invariant. With the denoising term $\hat{\boldsymbol{\epsilon}}_{\boldsymbol{F}}$ predicted by $\phi(\boldsymbol{L}_t, \boldsymbol{F}_t, \boldsymbol{A}, t)$, we combine the ancestral predictor [38, 50] with the Langevin corrector [49] to sample $\boldsymbol{F}_0$. We immediately have:

**Proposition 2.** *The marginal distribution $p(\boldsymbol{F}_0)$ is periodic translation invariant if $\hat{\boldsymbol{\epsilon}}_{\boldsymbol{F}}(\mathcal{M}_t, t)$ is periodic translation invariant, namely $\hat{\boldsymbol{\epsilon}}_{\boldsymbol{F}}(\boldsymbol{L}_t, \boldsymbol{F}_t, \boldsymbol{A}, t) = \hat{\boldsymbol{\epsilon}}_{\boldsymbol{F}}(\boldsymbol{L}_t, w(\boldsymbol{F}_t + \boldsymbol{t}\boldsymbol{1}^\top), \boldsymbol{A}, t), \forall \boldsymbol{t} \in \mathbb{R}^3$.*

The training objective for score matching is:

$$\mathcal{L}_{\boldsymbol{F}} = \mathbb{E}_{\boldsymbol{F}_t \sim q(\boldsymbol{F}_t|\boldsymbol{F}_0), t \sim \mathcal{U}(1,T)} \big[ \lambda_t \| \nabla_{\boldsymbol{F}_t} \log q(\boldsymbol{F}_t|\boldsymbol{F}_0) - \hat{\boldsymbol{\epsilon}}_{\boldsymbol{F}}(\mathcal{M}_t, t) \|_2^2 \big],$$

where $\lambda_t = \mathbb{E}_{\boldsymbol{F}_t}^{-1} \big[ \| \nabla_{\boldsymbol{F}_t} \log q(\boldsymbol{F}_t|\boldsymbol{F}_0) \|_2^2 \big]$ is approximated via Monte-Carlo sampling. More details are deferred to Appendix B.1.

**Extension to ab initio crystal generation** Although our method is proposed to address CSP where the composition $\boldsymbol{A}$ is fixed, our method is able to be extended for the ab initio generation task by further generating $\boldsymbol{A}$. We achieve this by additionally optimizing the one-hot representation $\boldsymbol{A}$ with a DDPM-based approach. We provide more details in Appendix G.

## 4.3 The Architecture of the Denoising Model

This subsection designs the denoising model $\phi(\boldsymbol{L}, \boldsymbol{F}, \boldsymbol{A}, t)$ that outputs $\hat{\boldsymbol{\epsilon}}_{\boldsymbol{L}}$ and $\hat{\boldsymbol{\epsilon}}_{\boldsymbol{F}}$ satisfying the properties stated in Proposition 1 and 2.

Let $\boldsymbol{H}^{(s)} = [\boldsymbol{h}_1^{(s)}, \cdots, \boldsymbol{h}_N^{(s)}]$ denote the node representations of the $s$-th layer. The input feature is given by $\boldsymbol{h}_i^{(0)} = \rho(f_{\text{atom}}(\boldsymbol{a}_i), f_{\text{pos}}(t))$, where $f_{\text{atom}}$ and $f_{\text{pos}}$ are the atomic embedding and sinusoidal positional encoding [38, 51], respectively; $\rho$ is a multi-layer perception (MLP).

Built upon EGNN [32], the $s$-th layer message-passing is unfolded as follows:

Here $\varphi_m$ and $\varphi_h$ are MLPs. The function $\psi_{\text{FT}} : (-1, 1)^3 \rightarrow [-1, 1]^{3 \times K}$ is Fourier Transformation of the relative fractional coordinate $\boldsymbol{f}_j - \boldsymbol{f}_i$. Specifically, suppose the input to be $\boldsymbol{f} = [f_1, f_2, f_3]^\top$, then the $c$-th row and $k$-th column of the output is calculated by

$$\boldsymbol{m}_{ij}^{(s)} = \varphi_m(\boldsymbol{h}_i^{(s-1)}, \boldsymbol{h}_j^{(s-1)}, \boldsymbol{L}^\top \boldsymbol{L}, \psi_{\text{FT}}(\boldsymbol{f}_j - \boldsymbol{f}_i)), \quad (6)$$

$$\boldsymbol{m}_i^{(s)} = \sum_{j=1}^N \boldsymbol{m}_{ij}^{(s)}, \quad (7)$$

$$\boldsymbol{h}_i^{(s)} = \boldsymbol{h}_i^{(s-1)} + \varphi_h(\boldsymbol{h}_i^{(s-1)}, \boldsymbol{m}_i^{(s)}). \quad (8)$$

$\psi_{\text{FT}}(\boldsymbol{f})[c, k] = \sin(2\pi m f_c)$, if $k = 2m$ (even); and $\psi_{\text{FT}}(\boldsymbol{f})[c, k] = \cos(2\pi m f_c)$, if $k = 2m + 1$ (odd). $\psi_{\text{FT}}$ extracts various frequencies of all relative fractional distances that are helpful for crystal structure modeling, and more importantly, $\psi_{\text{FT}}$ is periodic translation invariant, namely, $\psi_{\text{FT}}(w(\boldsymbol{f}_j + \boldsymbol{t}) - w(\boldsymbol{f}_i + \boldsymbol{t})) = \psi_{\text{FT}}(\boldsymbol{f}_j - \boldsymbol{f}_i)$ for any translation $\boldsymbol{t}$, which is proved in Appendix A.3.

After $S$ layers of message passing conducted on the fully connected graph, the lattice noise $\hat{\boldsymbol{\epsilon}}_{\boldsymbol{L}}$ is acquired by a linear combination of $\boldsymbol{L}$, with the weights given by the final layer:

$$\hat{\boldsymbol{\epsilon}}_{\boldsymbol{L}} = \boldsymbol{L} \varphi_L \Big( \frac{1}{N} \sum_{i=1}^N \boldsymbol{h}_i^{(S)} \Big), \quad (9)$$

where $\varphi_L$ is an MLP with output shape as $3 \times 3$. The fractional coordinate score $\hat{\boldsymbol{\epsilon}}_{\boldsymbol{F}}$ is output by:

$$\hat{\boldsymbol{\epsilon}}_{\boldsymbol{F}}[:, i] = \varphi_F(\boldsymbol{h}_i^{(S)}), \quad (10)$$

where $\hat{\boldsymbol{\epsilon}}_{\boldsymbol{F}}[:, i]$ defines the $i$-th column of $\hat{\boldsymbol{\epsilon}}_{\boldsymbol{F}}$, and $\varphi_F$ is an MLP on the final representation.

We apply the inner product term $\boldsymbol{L}^\top \boldsymbol{L}$ in Eq. (6) to achieve O(3)-invariance, as $(\boldsymbol{Q}\boldsymbol{L})^\top(\boldsymbol{Q}\boldsymbol{L}) = \boldsymbol{L}^\top \boldsymbol{L}$ for any orthogonal matrix $\boldsymbol{Q} \in \mathbb{R}^{3 \times 3}$. This leads to the O(3)-invariance of $\varphi_L$ in Eq. (10), and we further left-multiply $\boldsymbol{L}$ with $\varphi_L$ to ensure the O(3)-equivariance of $\hat{\boldsymbol{\epsilon}}_{\boldsymbol{L}}$. Therefore, the above formulation of the denoising model $\phi(\boldsymbol{L}, \boldsymbol{F}, \boldsymbol{A}, t)$ ensures the following property.

**Proposition 3.** *The score $\hat{\boldsymbol{\epsilon}}_{\boldsymbol{L}}$ by Eq. (9) is O(3)-equivariant, and the score $\hat{\boldsymbol{\epsilon}}_{\boldsymbol{F}}$ from Eq. (10) is periodic translation invariant. Hence, the generated distribution by DiffCSP is periodic E(3) invariant.*

**Comparison with multi-graph representation** Previous methods [9, 15, 29, 52] utilize Cartesian coordinates, and usually describe crystals with multi-graph representation to encode the periodic structures. They create multiple edges to connect each pair of nodes where different edges refer to different integral cell translations. Here, we no longer require multi-graph representation, since we employ fractional coordinates that naturally encode periodicity and the Fourier transform $\psi_{\text{FT}}$ in our message passing is already periodic translation invariant. We will ablate the benefit in Table 3.

Table 1: Results on stable structure prediction task.

| | # of samples | Perov-5 Match rate↑ | RMSE↓ | Carbon-24 Match rate↑ | RMSE↓ | MP-20 Match rate↑ | RMSE↓ | MPTS-52 Match rate↑ | RMSE↓ |
|---|---|---|---|---|---|---|---|---|---|
| RS [21] | 20 | 29.22 | 0.2924 | 14.63 | 0.4041 | 8.73 | 0.2501 | 2.05 | 0.3329 |
| | 5,000 | 36.56 | 0.0886 | 14.63 | 0.4041 | 11.49 | 0.2822 | 2.68 | 0.3444 |
| BO [21] | 20 | 21.03 | 0.2830 | 0.44 | 0.3653 | 8.11 | 0.2402 | 2.05 | 0.3024 |
| | 5,000 | 55.09 | 0.2037 | 12.17 | 0.4089 | 12.68 | 0.2816 | 6.69 | 0.3444 |
| PSO [21] | 20 | 20.90 | 0.0836 | 6.40 | 0.4204 | 4.05 | 0.1567 | 1.06 | 0.2339 |
| | 5,000 | 21.88 | 0.0844 | 6.50 | 0.4211 | 4.35 | 0.1670 | 1.09 | 0.2390 |
| P-cG-SchNet [53] | 1 | 48.22 | 0.4179 | 17.29 | 0.3846 | 15.39 | 0.3762 | 3.67 | 0.4115 |
| | 20 | 97.94 | 0.3463 | 55.91 | 0.3551 | 32.64 | 0.3018 | 12.96 | 0.3942 |
| CDVAE [9] | 1 | 45.31 | 0.1138 | 17.09 | 0.2969 | 33.90 | 0.1045 | 5.34 | 0.2106 |
| | 20 | 88.51 | 0.0464 | 88.37 | 0.2286 | 66.95 | 0.1026 | 20.79 | 0.2085 |
| DiffCSP | 1 | 52.02 | 0.0760 | 17.54 | 0.2759 | 51.49 | 0.0631 | 12.19 | 0.1786 |
| | 20 | **98.60** | **0.0128** | **88.47** | **0.2192** | **77.93** | **0.0492** | **34.02** | **0.1749** |

Figure 3: Visualization of the predicted structures from different methods. We select the structure of the lowest RMSE over 20 candidates. We translate the same predicted atom by all methods to the origin for better comparison. Our DiffCSP accurately delivers high quality structure predictions.

## 5 Experiments

In this section, we evaluate the efficacy of DiffCSP on a diverse range of tasks, by showing that it can generate high-quality structures of different crystals in § 5.1, with lower time cost comparing with DFT-based optimization method in § 5.2. Ablations in § 5.3 exhibit the necessity of each designed component. We further showcase the capability of DiffCSP in the ab initio generation task in § 5.4.

### 5.1 Stable Structure Prediction

**Dataset** We conduct experiments on four datasets with distinct levels of difficulty. **Perov-5** [54, 55] contains 18,928 perovskite materials with similar structures. Each structure has 5 atoms in a unit cell. **Carbon-24** [56] includes 10,153 carbon materials with 6∼24 atoms in a cell. **MP-20** [57] selects 45,231 stable inorganic materials from Material Projects [57], which includes the majority of experimentally-generated materials with at most 20 atoms in a unit cell. **MPTS-52** is a more challenging extension of MP-20, consisting of 40,476 structures up to 52 atoms per cell, sorted according to the earliest published year in literature. For Perov-5, Carbon-24 and MP-20, we apply the 60-20-20 split in line with Xie et al. [9]. For MPTS-52, we split 27,380/5,000/8,096 for training/validation/testing in chronological order.

**Baselines** We contrast two types of previous works. The first type follows the predict-optimize paradigm, which first trains a predictor of the target property and then utilizes certain optimization algorithms to search for optimal structures. Following Cheng et al. [21], we apply MEGNet [52] as the predictor of the formation energy. For the optimization algorithms, we choose Random Search (**RS**), Bayesian Optimization (**BO**), and Particle Swarm Optimization (**PSO**), all iterated over 5,000 steps. The second type is based on deep generative models. We follow the modification in Xie et al. [9], and leverage cG-SchNet [53] that utilizes SchNet [29] as the backbone and additionally

considers the ground-truth lattice initialization for encoding periodicity, yielding a final model named **P-cG-SchNet**. Another baseline **CDVAE** [9] is a VAE-based framework for pure crystal generation, by first predicting the lattice and the initial composition and then optimizing the atom types and coordinates via annealed Langevin dynamics [49]. To adapt CDVAE into the CSP task, we replace the original normal prior for generation with a parametric prior conditional on the encoding of the given composition. More details are provided in Appendix B.2.

**Evaluation metrics** Following the common practice [9], we evaluate by matching the predicted candidates with the ground-truth structure. Specifically, for each structure in the test set, we first generate $k$ samples of the same composition and then identify the matching if at least one of the samples matches the ground truth structure, under the metric by the StructureMatcher class in pymatgen [58] with thresholds stol=0.5, angle_tol=10, ltol=0.3. The **Match rate** is the proportion of the matched structures over the test set. **RMSE** is calculated between the ground truth and the best matching candidate, normalized by $\sqrt[3]{V/N}$ where $V$ is the volume of the lattice, and averaged over the matched structures. For optimization methods, we select 20 structures of the lowest energy of all 5,000 structures from all iterations during testing as candidates. For generative baselines and our DiffCSP, we let $k = 1$ and $k = 20$ for evaluation. We provide more details in Appendix B, C.1 and I.

**Results** Table 1 conveys the following observations. **1.** The optimization methods encounter low Match rates, signifying the difficulty of locating the optimal structures within the vast search space. **2.** In comparison to other generative methods that construct structures atom by atom or predict the lattice and atom coordinates in two stages, our method demonstrates superior performance, highlighting the effectiveness of jointly refining the lattice and coordinates during generation. **3.** All methods struggle with performance degradation as the number of atoms per cell increases, on the datasets from Perov-5 to MPTS-52. For example, the Match rates of the optimization methods are less than 10% in MPTS-52. Even so, our method consistently outperforms all other methods.

**Visualization** Figure 3 provides qualitative comparisons.DiffCSP clearly makes the best predictions.

## 5.2 Comparison with DFT-based Methods

We further select 10 binary and 5 ternary compounds in MP-20 testing set and compare our model with USPEX [59], a DFT-based software equipped with the evolutionary algorithm to search for stable structures.

Table 2: Overall results over the 15 selected compounds.

|  | Match rate (%)↑ | Avg. RMSD↓ | Avg. Time↓ |
|---|---|---|---|
| USPEX [59] | 53.33 | **0.0159** | 12.5h |
| DiffCSP | **73.33** | 0.0172 | **10s** |

For our method, we sample 20 candidates for each compound following the setting in Table 1. We select the model trained on MP-20 for inference, with a training duration of 5.2 hours. For USPEX, we apply 20 generations, 20 populations for each compound, and select the best sample in each generation, leading to 20 candidates as well. We summarize the **Match rate** over the 15 compounds, the **Averaged RMSD** over the matched structures, and the **Averaged Inference Time** to generate 20 candidates for each compound in Table 10. The detailed results for each compound are listed in Appendix F. DiffCSP correctly predicts more structures with higher match rate, and more importantly, its time cost is much less than USPEX, allowing more potential for real applications.

## 5.3 Ablation Studies

We ablate each component of DiffCSP in Table 3, and probe the following aspects. **1.** To verify the necessity of jointly updating the lattice $L$ and fractional coordinates $F$, we construct two variants that separate the joint optimization into two stages, denoted as $L \rightarrow F$ and $F \rightarrow L$. Particularly, $L \rightarrow F$ applies two networks to learn the reverse processes $p_{\theta_1}(L_{0:T-1}|A, F_T, L_T)$ and $p_{\theta_2}(F_{0:T-1}|A, F_T, L_0)$. During inference, we first sample $L_T, F_T$ from their prior distributions, acquiring $L_0$ via $p_{\theta_1}$, and then $F_0$ by $p_{\theta_2}$ based on $L_0$. $F \rightarrow L$ is similarly executed but with the generation order of $L_0$ and $F_0$ exchanged. Results indicate that $L \rightarrow F$ performs better than the $F \rightarrow L$, but both are inferior to the joint update in DiffCSP, which endorses our design. We conjecture that the joint diffusion fashion enables $L$ and $F$ to update synergistically, which makes the generation process more tractable to learn and thus leads to better performance. **2.** We explore the necessity of preserving the O(3) invariance when generating $L$, which is ensured by the inner product $L^\top L$ in Eq. (6).

When we replace it with $\boldsymbol{L}$ and change the final output as $\hat{\boldsymbol{\epsilon}}_{\boldsymbol{L}} = \varphi_L\left(\frac{1}{N}\sum_{i=1}^{N} \boldsymbol{h}_i^{(S)}\right)$ in Eq. (9) to break the equivariance, the model suffers from extreme performance detriment. Only 1.66% structures are successfully matched, which obviously implies the importance of incorporating $O(3)$ equivariance. Furthermore, we introduce the chirality into the denoising model by adding sign($|\boldsymbol{L}|$), the sign of the determinant of the lattice matrix, as an additional input in Eq. 6. The adapted model is $SO(3)$-invariant, but breaks the reflection symmetry and hence is NOT $O(3)$-invariant. There is no essential performance change, indicating the chirality is not quite crucial in distinguishing different crystal structures for the datasets used in this paper. **3.** We further assess the importance of periodic translation invariance from two perspectives. For the generation process, we generate $\boldsymbol{F}$ via the score-based

Table 3: Ablation studies on MP-20. MG: **M**ulti-**G**raph edge construction [15], FT: **F**ourier-**T**ransformation.

|  | Match rate (%) ↑ | RMSE ↓ |
|---|---|---|
| DiffCSP | **51.49** | **0.0631** |
| *w/o Joint Diffusion* | | |
| $\boldsymbol{L} \to \boldsymbol{F}$ | 50.03 | 0.0921 |
| $\boldsymbol{F} \to \boldsymbol{L}$ | 36.73 | 0.0838 |
| *w/o O(3) Equivariance* | | |
| *w/o* inner product | 1.66 | 0.4002 |
| *w/* chirality | 49.68 | 0.0637 |
| *w/o Periodic Translation Invariance* | | |
| *w/o* WN | 34.09 | 0.2350 |
| *w/o* FT | 29.15 | 0.0926 |
| *MG Edge Construction* | | |
| MG *w/* FT | 25.85 | 0.1079 |
| MG *w/o* FT | 28.05 | 0.1314 |

model with the Wrapped Normal (WN) distribution. We replace this module with DDPM under standard Gaussian as $q(\boldsymbol{F}_t|\mathcal{M}_0) = \mathcal{N}\left(\boldsymbol{F}_t|\sqrt{\bar{\alpha}_t}\boldsymbol{F}_0, (1-\bar{\alpha}_t)\boldsymbol{I}\right)$ similarly defined as Eq. (3). A lower match rate and higher RMSE are observed for this variant. For the model architecture, we adopt Fourier Transformation(FT) in Eq. (6) to capture periodicity. To investigate its effect, we replace $\psi_{\text{FT}}(\boldsymbol{f}_j - \boldsymbol{f}_i)$ with $\boldsymbol{f}_j - \boldsymbol{f}_i$, and the match rate drops from 51.49% to 29.15%. Both of the two observations verify the importance of retaining the periodic translation invariance. **4.** We further change the fully-connected graph into the multi-graph approach adopted in Xie and Grossman [15]. The multi-graph approach decreases the match rate, since the multi-graphs constructed under different intermediate structures may differ vibrantly during generation, leading to substantially higher training difficulty and lower sampling stability. We will provide more discussions in Appendix E.

## 5.4 Ab Initio Crystal Generation

DiffCSP is extendable to ab initio crystal generation by further conducting discrete diffusion on atom types $\boldsymbol{A}$. We contrast DiffCSP against five generative methods following [9]: **FTCP** [28], **Cond-DFC-VAE** [7], **G-SchNet** [60] and its periodic variant **P-G-SchNet**, and the orginal version of **CDVAE** [9]. Specifically for our DiffCSP, we gather the statistics of the atom numbers from the training set, then sample the number based on the pre-computed distribution similar to Hoogeboom et al. [41], which allows DiffCSP to generate structures of variable size. Following [9], we evaluate the generation performance in terms of there metrics: **Validity**, **Coverage**, and **Property statistics**, which respectively return the validity of the predicted crystals, the similarity between the test set and the generated samples, and the property calculation regarding density, formation energy, and the number of elements. The detailed definitions of the above metrics are provided in Appendix G.

**Results** Table 4 show that our method achieves comparable validity and coverage rate with previous methods, and significantly outperforms the baselines on the similarity of property statistics, which indicates the high reliability of the generated samples.

## 6 Discussions

**Limitation 1.** Composition generation. Our model yields slightly lower compositional validity in Table 4. We provide more discussion in Appendix G, and it is promising to propose more powerful generation methods on atom types. **2.** Experimental evaluation. Further wet-lab experiments can better verify the effectiveness of the model in real applications.

**Conclusion** In this work, we present DiffCSP, a diffusion-based learning framework for crystal structure prediction, which is particularly curated with the vital symmetries existing in crystals. The diffusion is highly flexible by jointly optimizing the lattice and fractional coordinates, where

---

[3] Composition-based metrics are not meaningful for Carbon-24, as all structures are composed of carbon.

Table 4: Results on ab initio generation task. The results of baseline methods are from Xie et al. [9].

| Data | Method | Validity (%) ↑ | | Coverage (%) ↑ | | Property ↓ | | |
|------|--------|------|------|-------|-------|---------|---------|-----------------|
| | | Struc. | Comp. | COV-R | COV-P | $d_\rho$ | $d_E$ | $d_{\text{elem}}$ |
| Perov-5 | FTCP [28] | 0.24 | 54.24 | 0.00 | 0.00 | 10.27 | 156.0 | 0.6297 |
| | Cond-DFC-VAE [7] | 73.60 | 82.95 | 73.92 | 10.13 | 2.268 | 4.111 | 0.8373 |
| | G-SchNet [60] | 99.92 | 98.79 | 0.18 | 0.23 | 1.625 | 4.746 | 0.0368 |
| | P-G-SchNet [60] | 79.63 | **99.13** | 0.37 | 0.25 | 0.2755 | 1.388 | 0.4552 |
| | CDVAE [9] | **100.0** | 98.59 | 99.45 | **98.46** | 0.1258 | 0.0264 | 0.0628 |
| | DiffCSP | **100.0** | 98.85 | **99.74** | 98.27 | **0.1110** | **0.0263** | **0.0128** |
| Carbon-24[3] | FTCP [28] | 0.08 | – | 0.00 | 0.00 | 5.206 | 19.05 | – |
| | G-SchNet [60] | 99.94 | – | 0.00 | 0.00 | 0.9427 | 1.320 | – |
| | P-G-SchNet [60] | 48.39 | – | 0.00 | 0.00 | 1.533 | 134.7 | – |
| | CDVAE [9] | **100.0** | – | 99.80 | 83.08 | 0.1407 | 0.2850 | – |
| | DiffCSP | **100.0** | – | **99.90** | **97.27** | **0.0805** | **0.0820** | – |
| MP-20 | FTCP [28] | 1.55 | 48.37 | 4.72 | 0.09 | 23.71 | 160.9 | 0.7363 |
| | G-SchNet [60] | 99.65 | 75.96 | 38.33 | 99.57 | 3.034 | 42.09 | 0.6411 |
| | P-G-SchNet [60] | 77.51 | 76.40 | 41.93 | 99.74 | 4.04 | 2.448 | 0.6234 |
| | CDVAE [9] | **100.0** | **86.70** | 99.15 | 99.49 | 0.6875 | 0.2778 | 1.432 |
| | DiffCSP | **100.0** | 83.25 | **99.71** | **99.76** | **0.3502** | **0.1247** | **0.3398** |

the intermediate distributions are guaranteed to be invariant under necessary transformations. We demonstrate the efficacy of our approach on a wide range of crystal datasets, verifying the strong applicability of DiffCSP towards predicting high-quality crystal structures.

# 7 Acknowledgement

This work was jointly supported by the following projects: the National Natural Science Foundation of China (61925601 & 62006137); Beijing Nova Program (20230484278); the Fundamental Research Funds for the Central Universities, and the Research Funds of Renmin University of China (23XNKJ19); Alibaba Damo Research Fund; CCF-Ant Research Fund (CCF-AFSG RF20220204); National Key R&D Program of China (2022ZD0117805).

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

# Contents

# A  Theoretical Analysis

## A.1  Proof of Proposition 1

We first introduce the following definition to describe the equivariance and invariance from the perspective of distributions.

**Definition 4.** *We call a distribution $p(x)$ is G-invariant if for any transformation $g$ in the group $G$, $p(g \cdot x) = p(x)$, and a conditional distribution $p(x|c)$ is G-equivariant if $p(g \cdot x|g \cdot c) = p(x|c), \forall g \in G$.*

We then provide and prove the following lemma to capture the symmetry of the generation process.

**Lemma 1** (Xu et al. [10])**.** *Consider the generation Markov process $p(x_0) = p(x_T) \int p(x_{0:T-1}|x_t)dx_{1:T}$. If the prior distribution $p(x_T)$ is G-invariant and the Markov transitions $p(x_{t-1}|x_t), 0 < t \leq T$ are G-equivariant, the marginal distribution $p(x_0)$ is also G-invariant.*

*Proof.* For any $g \in G$, we have

$$p(g \cdot x_0) = p(g \cdot x_T) \int p(g \cdot x_{0:T-1}|g \cdot x_t)dx_{1:T}$$

$$= p(g \cdot x_T) \int \prod_{t=1}^{T} p(g \cdot x_{t-1}|g \cdot x_t)dx_{1:T}$$

$$= p(x_T) \int \prod_{t=1}^{T} p(g \cdot x_{t-1}|g \cdot x_t)dx_{1:T}$$

$$= p(x_T) \int \prod_{t=1}^{T} p(x_{t-1}|x_t)dx_{1:T}$$

$$= p(x_T) \int p(x_{0:T-1}|x_t)dx_{1:T}$$

$$= p(x_0).$$

Hence, the marginal distribution $p(x_0)$ is G-invariant. $\square$

The proposition Proposition 1 is rewritten and proved as follows.

**Proposition 1.** *The marginal distribution $p(\boldsymbol{L}_0)$ by Eq. (3) is O(3)-invariant if $\hat{\boldsymbol{\epsilon}}_{\boldsymbol{L}}(\mathcal{M}_t, t)$ is O(3)-equivariant, namely $\hat{\boldsymbol{\epsilon}}_{\boldsymbol{L}}(\boldsymbol{Q}\boldsymbol{L}_t, \boldsymbol{F}_t, \boldsymbol{A}, t) = \boldsymbol{Q}\hat{\boldsymbol{\epsilon}}_{\boldsymbol{L}}(\boldsymbol{L}_t, \boldsymbol{F}_t, \boldsymbol{A}, t), \forall \boldsymbol{Q}^\top \boldsymbol{Q} = \boldsymbol{I}$.*

*Proof.* Consider the transition probability in Eq. (3), we have

$$p(\boldsymbol{L}_{t-1}|\boldsymbol{L}_t, \boldsymbol{F}_t, \boldsymbol{A}) = \mathcal{N}(\boldsymbol{L}_{t-1}|a_t(\boldsymbol{L}_t - b_t\hat{\boldsymbol{\epsilon}}_{\boldsymbol{L}}(\boldsymbol{L}_t, \boldsymbol{F}_t, \boldsymbol{A}, t)), \sigma_t^2 \boldsymbol{I}),$$

where $a_t = \frac{1}{\sqrt{\alpha_t}}, b_t = \frac{\beta_t}{\sqrt{1-\bar{\alpha}_t}}, \sigma_t^2 = \beta_t \cdot \frac{1-\bar{\alpha}_{t-1}}{1-\bar{\alpha}_t}$ for simplicity, and $\hat{\boldsymbol{\epsilon}}_{\boldsymbol{L}}(\mathcal{M}_t, t)$ is completed as $\hat{\boldsymbol{\epsilon}}_{\boldsymbol{L}}(\boldsymbol{L}_t, \boldsymbol{F}_t, \boldsymbol{A}, t)$.

As the denoising term $\hat{\boldsymbol{\epsilon}}_{\boldsymbol{L}}(\boldsymbol{L}_t, \boldsymbol{F}_t, \boldsymbol{A}, t)$ is O(3)-equivariant, we have $\hat{\boldsymbol{\epsilon}}_{\boldsymbol{L}}(\boldsymbol{Q}\boldsymbol{L}_t, \boldsymbol{F}_t, \boldsymbol{A}, t) = \boldsymbol{Q}\hat{\boldsymbol{\epsilon}}_{\boldsymbol{L}}(\boldsymbol{L}_t, \boldsymbol{F}_t, \boldsymbol{A}, t)$ for any orthogonal transformation $\boldsymbol{Q} \in \mathbb{R}^{3\times3}, \boldsymbol{Q}^\top \boldsymbol{Q} = \boldsymbol{I}$.

For the variable $\boldsymbol{L} \sim \mathcal{N}(\bar{\boldsymbol{L}}, \sigma^2 \boldsymbol{I})$, we have $\boldsymbol{Q}\boldsymbol{L} \sim \mathcal{N}(\boldsymbol{Q}\bar{\boldsymbol{L}}, \boldsymbol{Q}(\sigma^2 \boldsymbol{I})\boldsymbol{Q}^\top) = \mathcal{N}(\boldsymbol{Q}\bar{\boldsymbol{L}}, \sigma^2 \boldsymbol{I})$. That is,

$$\mathcal{N}(\boldsymbol{L}|\bar{\boldsymbol{L}}, \sigma^2 \boldsymbol{I}) = \mathcal{N}(\boldsymbol{Q}\boldsymbol{L}|\boldsymbol{Q}\bar{\boldsymbol{L}}, \sigma^2 \boldsymbol{I}). \tag{11}$$

For the transition probability $p(\boldsymbol{L}_{t-1}|\boldsymbol{L}_t, \boldsymbol{F}_t, \boldsymbol{A})$, we have

$$
\begin{aligned}
p(\boldsymbol{Q}\boldsymbol{L}_{t-1}|\boldsymbol{Q}\boldsymbol{L}_t, \boldsymbol{F}_t, \boldsymbol{A}) &= \mathcal{N}(\boldsymbol{Q}\boldsymbol{L}_{t-1}|a_t(\boldsymbol{Q}\boldsymbol{L}_t - b_t\hat{\boldsymbol{\epsilon}}_{\boldsymbol{L}}(\boldsymbol{Q}\boldsymbol{L}_t, \boldsymbol{F}_t, \boldsymbol{A}, t)), \sigma_t^2\boldsymbol{I}) \\
&= \mathcal{N}(\boldsymbol{Q}\boldsymbol{L}_{t-1}|a_t(\boldsymbol{Q}\boldsymbol{L}_t - b_t\boldsymbol{Q}\hat{\boldsymbol{\epsilon}}_{\boldsymbol{L}}(\boldsymbol{L}_t, \boldsymbol{F}_t, \boldsymbol{A}, t)), \sigma_t^2\boldsymbol{I}) \\
&\qquad\qquad\qquad\qquad\qquad\qquad\qquad\quad (\text{O}(3)\text{-equivariant } \hat{\boldsymbol{\epsilon}}_{\boldsymbol{L}}) \\
&= \mathcal{N}\Big(\boldsymbol{Q}\boldsymbol{L}_{t-1}\Big|\boldsymbol{Q}\Big(a_t(\boldsymbol{L}_t - b_t\hat{\boldsymbol{\epsilon}}_{\boldsymbol{L}}(\boldsymbol{L}_t, \boldsymbol{F}_t, \boldsymbol{A}, t))\Big), \sigma_t^2\boldsymbol{I}\Big) \\
&= \mathcal{N}(\boldsymbol{L}_{t-1}|a_t(\boldsymbol{L}_t - b_t\hat{\boldsymbol{\epsilon}}_{\boldsymbol{L}}(\boldsymbol{L}_t, \boldsymbol{F}_t, \boldsymbol{A}, t)), \sigma_t^2\boldsymbol{I}) \qquad (\text{Eq. (11)}) \\
&= p(\boldsymbol{L}_{t-1}|\boldsymbol{L}_t, \boldsymbol{F}_t, \boldsymbol{A}).
\end{aligned}
$$

As the transition is $O(3)$-equivariant and the prior distribution $\mathcal{N}(0, \boldsymbol{I})$ is $O(3)$-invariant, we prove that the the marginal distribution $p(\boldsymbol{L}_0)$ is $O(3)$-invariant based on lemma 1. □

### A.2 Proof of Proposition 2

Let $\mathcal{N}_w(\mu, \sigma^2\boldsymbol{I})$ denote the wrapped normal distribution with mean $\mu$, variance $\sigma^2$ and period 1. We first provide the following lemma.

**Lemma 3.** *If the denoising term $\hat{\boldsymbol{\epsilon}}_{\boldsymbol{F}}(\boldsymbol{L}_t, \boldsymbol{F}_t, \boldsymbol{A}, t)$ is periodic translation invariant, and the transition probabilty can be formulated as $p(\boldsymbol{F}_{t-1}|\boldsymbol{L}_t, \boldsymbol{F}_t, \boldsymbol{A}) = \mathcal{N}_w(\boldsymbol{F}_{t-1}|\boldsymbol{F}_t + u_t\hat{\boldsymbol{\epsilon}}_{\boldsymbol{F}}(\boldsymbol{L}_t, \boldsymbol{F}_t, \boldsymbol{A}, t), v_t^2\boldsymbol{I})$, where $u_t, v_t$ are functions of $t$, the transition is periodic translation equivariant.*

*Proof.* As the denoising term $\hat{\boldsymbol{\epsilon}}_{\boldsymbol{F}}(\boldsymbol{L}_t, \boldsymbol{F}_t, \boldsymbol{A}, t)$ is periodic translation invariant (for short PTI), we have $\hat{\boldsymbol{\epsilon}}_{\boldsymbol{F}}(\boldsymbol{L}_t, w(\boldsymbol{F}_t + \boldsymbol{t}\boldsymbol{1}^\top), \boldsymbol{A}, t) = \hat{\boldsymbol{\epsilon}}_{\boldsymbol{F}}(\boldsymbol{L}_t, \boldsymbol{F}_t, \boldsymbol{A}, t)$, for any translation $\boldsymbol{t} \in \mathbb{R}^3$.

For the wrapping function $w(\cdot)$, we have

$$
w(a + b) = w(w(a) + b), \forall a, b \in \mathbb{R}. \tag{12}
$$

For wrapped normal distribution $\mathcal{N}_w(\mu, \sigma^2)$ with mean $\mu$, variance $\sigma^2$ and period 1, and for any $k', k'' \in \mathbb{Z}$, we have

$$
\begin{aligned}
\mathcal{N}_w(x + k'|\mu + k'', \sigma^2) &= \frac{1}{\sqrt{2\pi}\sigma} \sum_{k=-\infty}^{\infty} \exp\Big(-\frac{(x + k' - (\mu + k'') - k)^2}{2\sigma^2}\Big) \\
&= \frac{1}{\sqrt{2\pi}\sigma} \sum_{m=-\infty}^{\infty} \exp\Big(-\frac{(x - \mu - m)^2}{2\sigma^2}\Big) \qquad (m = k - k' + k'') \\
&= \mathcal{N}_w(x|\mu, \sigma^2)
\end{aligned}
$$

Let $k' = 0, k'' = w(\mu) - \mu$, we directly have

$$
\mathcal{N}_w(x|w(\mu), \sigma^2) = \mathcal{N}_w(x|\mu, \sigma^2). \tag{13}
$$

For any $t \in \mathbb{R}$, we have

$$
\begin{aligned}
\mathcal{N}_w(x + t|\mu + t, \sigma^2) &= \frac{1}{\sqrt{2\pi}\sigma} \sum_{k=-\infty}^{\infty} \exp\Big(-\frac{(x + t - (\mu + t) - k)^2}{2\sigma^2}\Big) \\
&= \frac{1}{\sqrt{2\pi}\sigma} \sum_{k=-\infty}^{\infty} \exp\Big(-\frac{(x - \mu - k)^2}{2\sigma^2}\Big) \\
&= \mathcal{N}_w(x|\mu, \sigma^2).
\end{aligned}
$$

Let $k' = w(x + t) - (x + t), k'' = w(\mu + t) - (\mu + t)$, we have

$$
\mathcal{N}_w(w(x + t)|w(\mu + t), \sigma^2) = \mathcal{N}_w(x + t|\mu + t, \sigma^2) = \mathcal{N}_w(x|\mu, \sigma^2). \tag{14}
$$

For the transition probability $p(\boldsymbol{F}_{t-1}|\boldsymbol{L}_t, \boldsymbol{F}_t, \boldsymbol{A})$, we have

$$
\begin{aligned}
&p(w(\boldsymbol{F}_{t-1} + \boldsymbol{t})|\boldsymbol{L}_t, w(\boldsymbol{F}_t + \boldsymbol{t}\boldsymbol{1}^\top), \boldsymbol{A}) \\
&= \mathcal{N}_w(w(\boldsymbol{F}_{t-1} + \boldsymbol{t}\boldsymbol{1}^\top)|w(\boldsymbol{F}_t + \boldsymbol{t}\boldsymbol{1}^\top) + u_t\hat{\boldsymbol{\epsilon}}_{\boldsymbol{F}}(\boldsymbol{L}_t, w(\boldsymbol{F}_t + \boldsymbol{t}\boldsymbol{1}^\top), \boldsymbol{A}, t), v_t^2\boldsymbol{I}) \\
&= \mathcal{N}_w(w(\boldsymbol{F}_{t-1} + \boldsymbol{t}\boldsymbol{1}^\top)|w(\boldsymbol{F}_t + \boldsymbol{t}\boldsymbol{1}^\top) + u_t\hat{\boldsymbol{\epsilon}}_{\boldsymbol{F}}(\boldsymbol{L}_t, \boldsymbol{F}_t, \boldsymbol{A}, t), v_t^2\boldsymbol{I}) &&\text{(PTI } \hat{\boldsymbol{\epsilon}}_{\boldsymbol{F}}\text{)} \\
&= \mathcal{N}_w(w(\boldsymbol{F}_{t-1} + \boldsymbol{t}\boldsymbol{1}^\top)|w\Big(w(\boldsymbol{F}_t + \boldsymbol{t}\boldsymbol{1}^\top) + u_t\hat{\boldsymbol{\epsilon}}_{\boldsymbol{F}}(\boldsymbol{L}_t, \boldsymbol{F}_t, \boldsymbol{A}, t)\Big), v_t^2\boldsymbol{I}) &&\text{(Eq. (13))} \\
&= \mathcal{N}_w(w(\boldsymbol{F}_{t-1} + \boldsymbol{t}\boldsymbol{1}^\top)|w\Big(\boldsymbol{F}_t + u_t\hat{\boldsymbol{\epsilon}}_{\boldsymbol{F}}(\boldsymbol{L}_t, \boldsymbol{F}_t, \boldsymbol{A}, t) + \boldsymbol{t}\Big), v_t^2\boldsymbol{I}) &&\text{(Eq. (12))} \\
&= \mathcal{N}_w(\boldsymbol{F}_{t-1}|\boldsymbol{F}_t + u_t\hat{\boldsymbol{\epsilon}}_{\boldsymbol{F}}(\boldsymbol{L}_t, \boldsymbol{F}_t, \boldsymbol{A}, t), v_t^2\boldsymbol{I}) &&\text{(Eq. (14))} \\
&= p(\boldsymbol{F}_{t-1}|\boldsymbol{L}_t, \boldsymbol{F}_t, \boldsymbol{A}).
\end{aligned}
$$

$\square$

The transition probability of the fractional coordinates during the Predictor-Corrector sampling can be formulated as

$$
p(\boldsymbol{F}_{t-1}|\boldsymbol{L}_t, \boldsymbol{F}_t, \boldsymbol{A}) = p_P(\boldsymbol{F}_{t-\frac{1}{2}}|\boldsymbol{L}_t, \boldsymbol{F}_t, \boldsymbol{A})p_C(\boldsymbol{F}_{t-1}|\boldsymbol{L}_{t-1}, \boldsymbol{F}_{t-\frac{1}{2}}, \boldsymbol{A}),
$$

$$
p_P(\boldsymbol{F}_{t-\frac{1}{2}}|\boldsymbol{L}_t, \boldsymbol{F}_t, \boldsymbol{A}) = \mathcal{N}_w(\boldsymbol{F}_{t-\frac{1}{2}}|\boldsymbol{F}_t + (\sigma_t^2 - \sigma_{t-1}^2)\hat{\boldsymbol{\epsilon}}_{\boldsymbol{F}}(\boldsymbol{L}_t, \boldsymbol{F}_t, \boldsymbol{A}, t), \frac{\sigma_{t-1}^2(\sigma_t^2 - \sigma_{t-1}^2)}{\sigma_t^2}\boldsymbol{I}),
$$

$$
p_C(\boldsymbol{F}_{t-1}|\boldsymbol{L}_{t-1}, \boldsymbol{F}_{t-\frac{1}{2}}, \boldsymbol{A}) = \mathcal{N}_w(\boldsymbol{F}_{t-\frac{1}{2}}|\boldsymbol{F}_t + \gamma\frac{\sigma_{t-1}}{\sigma_1}\hat{\boldsymbol{\epsilon}}_{\boldsymbol{F}}(\boldsymbol{L}_{t-1}, \boldsymbol{F}_{t-\frac{1}{2}}, \boldsymbol{A}, t-1), 2\gamma\frac{\sigma_{t-1}}{\sigma_1}\boldsymbol{I}),
$$

where $p_P, p_C$ are the transitions of the predictor and corrector. According to lemma 3, both of the transitions are periodic translation equivariant. Therefore, the transition $p(\boldsymbol{F}_{t-1}|\boldsymbol{L}_t, \boldsymbol{F}_t, \boldsymbol{A})$ is periodic translation equivariant. As the prior distribution $\mathcal{U}(0,1)$ is periodic translation invariant, we finally prove that the marginal distribution $p(\boldsymbol{F}_0)$ is periodic translation invariant based on lemma 1.

### A.3 Proof of Proposition 3

We rewrite proposition 3 as follows.

**Proposition 3.** *The score $\hat{\boldsymbol{\epsilon}}_{\boldsymbol{L}}$ by Eq. (9) is O(3)-equivariant, and the score $\hat{\boldsymbol{\epsilon}}_{\boldsymbol{F}}$ from Eq. (10) is periodic translation invariant. Hence, the generated distribution by DiffCSP is periodic E(3) invariant.*

*Proof.* We first prove the orthogonal invariance of the inner product term $\boldsymbol{L}^\top\boldsymbol{L}$. For any orthogonal transformation $\boldsymbol{Q} \in \mathbb{R}^{3\times3}, \boldsymbol{Q}^\top\boldsymbol{Q} = \boldsymbol{I}$, we have

$$
(\boldsymbol{Q}\boldsymbol{L})^\top(\boldsymbol{Q}\boldsymbol{L}) = \boldsymbol{L}^\top\boldsymbol{Q}^\top\boldsymbol{Q}\boldsymbol{L} = \boldsymbol{L}^\top\boldsymbol{I}\boldsymbol{L} = \boldsymbol{L}^\top\boldsymbol{L}.
$$

For the Fourier Transformation, consider $k$ is even, we have

$$
\begin{aligned}
&\psi_{\text{FT}}(w(\boldsymbol{f}_j + \boldsymbol{t}) - w(\boldsymbol{f}_i + \boldsymbol{t}))[c, k] \\
&= \sin\Big(2\pi m\big(w(f_{j,c} + t_c) - w(f_{i,c} + t_c)\big)\Big) \\
&= \sin\Big(2\pi m(f_{j,c} - f_{i,c}) - 2\pi m\big((f_{j,c} - f_{i,c}) - (w(f_{j,c} + t_c) - w(f_{i,c} + t_c))\big)\Big) \\
&= \sin(2\pi m(f_{j,c} - f_{i,c})) \\
&= \psi_{\text{FT}}(\boldsymbol{f}_j - \boldsymbol{f}_i)[c, k].
\end{aligned}
$$

Similar results can be acquired as $k$ is odd. Therefore, we have $\psi_{\text{FT}}(w(\boldsymbol{f}_j + \boldsymbol{t}) - w(\boldsymbol{f}_i + \boldsymbol{t})) = \psi_{\text{FT}}(\boldsymbol{f}_j - \boldsymbol{f}_i), \forall \boldsymbol{t} \in \mathbb{R}^3$, *i.e.*, the Fourier Transformation $\psi_{\text{FT}}$ is periodic translation invariant. According to the above, the message passing layers defined in Eq. (6)- (8) is periodic E(3) invariant. Hence, we can directly prove that the coordinate denoising term $\hat{\boldsymbol{\epsilon}}_{\boldsymbol{F}}$ is periodic translation invariant. Let $\hat{\boldsymbol{\epsilon}}_l(\boldsymbol{L}, \boldsymbol{F}, \boldsymbol{A}, t) = \varphi_L\big(\frac{1}{N}\sum_{i=1}^N \boldsymbol{h}_i^{(S)}\big)$. For the lattice denoising term $\hat{\boldsymbol{\epsilon}}_{\boldsymbol{L}} = \boldsymbol{L}\hat{\boldsymbol{\epsilon}}_l$, we have

$$
\begin{aligned}
\hat{\boldsymbol{\epsilon}}_{\boldsymbol{L}}(\boldsymbol{Q}\boldsymbol{L}, \boldsymbol{F}, \boldsymbol{A}, t) &= \boldsymbol{Q}\boldsymbol{L}\hat{\boldsymbol{\epsilon}}_l(\boldsymbol{Q}\boldsymbol{L}, \boldsymbol{F}, \boldsymbol{A}, t) \\
&= \boldsymbol{Q}\boldsymbol{L}\hat{\boldsymbol{\epsilon}}_l(\boldsymbol{L}, \boldsymbol{F}, \boldsymbol{A}, t) \\
&= \boldsymbol{Q}\hat{\boldsymbol{\epsilon}}_{\boldsymbol{L}}(\boldsymbol{L}, \boldsymbol{F}, \boldsymbol{A}, t), \forall \boldsymbol{Q} \in \mathbb{R}^{3\times3}, \boldsymbol{Q}^\top\boldsymbol{Q} = \boldsymbol{I}.
\end{aligned}
$$

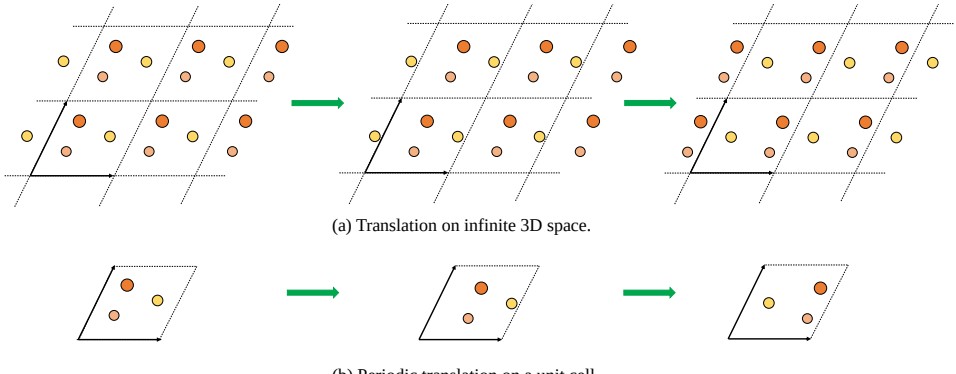

(a) Translation on infinite 3D space.

(b) Periodic translation on a unit cell.

Figure 4: An example of periodic translation invariance. From the view of a unit cell, the atoms translated across the right boundary will be brought back to the left side.

Above all, $\hat{\epsilon}_L$ is O(3)-equivariant, and $\hat{\epsilon}_F$ is periodic translation invariant. According to proposition 1 and 2, the generated distribution by DiffCSP in Algorithm 2 is periodic E(3) invariant. $\qquad \square$

### A.4 Discussion on Periodic Translation Invariance

In Definition 3, we define the periodic translation invariance as a combination of translation invariance and periodicity. To see this, we illustrate an additional example in Figure 4. From a global view, when we translate all atom coordinates from left to right, the crystal structure remains unchanged, which indicates translation invariance. At the same time, from the view of a unit cell, the atom translated across the right boundary will be brought back to the left side owing to periodicity. Therefore, for convenience, we define the joint effect of translation invariance and periodicity as periodic translation invariance.

Previous works [36] have shown that shifting the periodic boundaries will not change the crystal structure. In this section, we further show that such periodic boundary shifting is equivalent to the periodic translation defined in Definition 3.

Consider two origin points $\boldsymbol{p}_1, \boldsymbol{p}_2 \in \mathbb{R}^{3 \times 1}$ and the lattice matrix $\boldsymbol{L}$, the constructed unit cells by $\boldsymbol{p}_1, \boldsymbol{p}_2$ can be represented as $\mathcal{M}_1 = (\boldsymbol{A}_1, \boldsymbol{F}_1, \boldsymbol{L})$ and $\mathcal{M}_2 = (\boldsymbol{A}_2, \boldsymbol{F}_2, \boldsymbol{L})$, where $\boldsymbol{F}_1, \boldsymbol{F}_2 \in \mathbb{R}^{3 \times N}$ are fractional coordinates and

$$\{(\boldsymbol{a}'_{1,i}, \boldsymbol{x}'_{1,i}) | \boldsymbol{a}'_{1,i} = \boldsymbol{a}_{1,i}, \boldsymbol{x}'_{1,i} = \boldsymbol{p}_1 + \boldsymbol{L}\boldsymbol{f}_{1,i} + \boldsymbol{L}\boldsymbol{k}, \forall \boldsymbol{k} \in \mathbb{Z}^{3 \times 1}\} \tag{15}$$

$$=\{(\boldsymbol{a}'_{1,j}, \boldsymbol{x}'_{1,j}) | \boldsymbol{a}'_{1,j} = \boldsymbol{a}_{1,j}, \boldsymbol{x}'_{2,j} = \boldsymbol{p}_2 + \boldsymbol{L}\boldsymbol{f}_{2,j} + \boldsymbol{L}\boldsymbol{k}, \forall \boldsymbol{k} \in \mathbb{Z}^{3 \times 1}\}, \tag{16}$$

which means that the unit cells formed by different origin points actually represent the same infinite crystal structures [36]. We further construct a bijection $\mathcal{T} : \mathcal{M}_1 \to \mathcal{M}_2$ mapping each atom in $\mathcal{M}_1$ to the corresponding atom in unit cell $\mathcal{M}_2$. For the pair $(a_{1,i}, f_{1,i}) \in \mathcal{M}_1, (a_{2,j}, f_{2,j}) \in \mathcal{M}_2$, we have $\mathcal{T}(a_{1,i}, f_{1,i}) = (a_{2,j}, f_{2,j})$ iff $\exists \boldsymbol{k}_i \in \mathbb{Z}^{3 \times 1}$, s.t.

$$\begin{cases} \boldsymbol{a}_{1,i} = \boldsymbol{a}_{2,j}, \\ \boldsymbol{p}_1 + \boldsymbol{L}\boldsymbol{f}_{1,i} + \boldsymbol{L}\boldsymbol{k}_i = \boldsymbol{p}_2 + \boldsymbol{L}\boldsymbol{f}_{2,j}. \end{cases}$$

After proper transformation, we have

$$\boldsymbol{f}_{2,j} = \boldsymbol{f}_{1,i} + \boldsymbol{L}^{-1}(\boldsymbol{p}_1 - \boldsymbol{p}_2) + \boldsymbol{k}_i. \tag{17}$$

As $\boldsymbol{f}_{1,i}, \boldsymbol{f}_{2,i} \in [0, 1)^{3 \times 1}$, we have

$$\begin{cases} \boldsymbol{k}_i = -\lfloor \boldsymbol{f}_{1,i} + \boldsymbol{L}^{-1}(\boldsymbol{p}_1 - \boldsymbol{p}_2) \rfloor, \\ \boldsymbol{f}_{2,j} = w\Big( \boldsymbol{f}_{1,i} + \boldsymbol{L}^{-1}(\boldsymbol{p}_1 - \boldsymbol{p}_2) \Big), \end{cases}$$

which means shifting the periodic boundaries by changing the origin point $\boldsymbol{p}_1$ into $\boldsymbol{p}_2$ is equivalent to a periodic translation $\boldsymbol{F}_2 = w\Big( \boldsymbol{F}_1 + \boldsymbol{L}^{-1}(\boldsymbol{p}_1 - \boldsymbol{p}_2)\boldsymbol{1}^\top \Big)$.

# B Implementation Details

## B.1 Approximation of the Wrapped Normal Distribution

The Probability Density Function (PDF) of the wrapped normal distribution $\mathcal{N}_w(0, \sigma_t^2)$ is

$$\mathcal{N}_w(x|0, \sigma_t^2) = \frac{1}{\sqrt{2\pi}\sigma_t} \sum_{k=-\infty}^{\infty} \exp\left(-\frac{(x-k)^2}{2\sigma_t^2}\right),$$

where $x \in [0, 1)$. Because the above series is convergent, it is reasonable to approximate the infinite summation to a finite truncated summation [61] as

$$f_{w,n}(x; 0, \sigma_t^2) = \frac{1}{\sqrt{2\pi}\sigma_t} \sum_{k=-n}^{n} \exp\left(-\frac{(x-k)^2}{2\sigma_t^2}\right).$$

And the logarithmic gradient of $f$ can be formulated as

$$\nabla_x \log f_{w,n}(x; 0, \sigma_t^2) = \nabla_x \log\left(\frac{1}{\sqrt{2\pi}\sigma_t} \sum_{k=-n}^{n} \exp\left(-\frac{(x-k)^2}{2\sigma_t^2}\right)\right)$$

$$= \nabla_x \log\left(\sum_{k=-n}^{n} \exp\left(-\frac{(x-k)^2}{2\sigma_t^2}\right)\right)$$

$$= \frac{\sum_{k=-n}^{n}(k-x)\exp\left(-\frac{(x-k)^2}{2\sigma_t^2}\right)}{\sigma_t^2 \sum_{k=-n}^{n}\exp\left(-\frac{(x-k)^2}{2\sigma_t^2}\right)}$$

To estimate $\lambda_t = \mathbb{E}_{x \sim \mathcal{N}_w(0,\sigma_t^2)}^{-1}\left[\|\nabla_x \log \mathcal{N}_w(x|0, \sigma_t^2)\|_2^2\right]$, we first sample $m$ points from $\mathcal{N}_w(0, \sigma_t^2)$, and the expectation is approximated as

$$\tilde{\lambda}_t = \left[\frac{1}{m}\sum_{i=1}^{m}\|\nabla_x \log f_{w,n}(x_i; 0, \sigma_t^2)\|_2^2\right]^{-1}$$

$$= \left[\frac{1}{m}\sum_{i=1}^{m}\left\|\frac{\sum_{k=-n}^{n}(k-x_i)\exp\left(-\frac{(x_i-k)^2}{2\sigma_t^2}\right)}{\sigma_t^2 \sum_{k=-n}^{n}\exp\left(-\frac{(x_i-k)^2}{2\sigma_t^2}\right)}\right\|_2^2\right]^{-1}.$$

For implementation, we select $n = 10$ and $m = 10000$.

## B.2 Adaptation of CDVAE

As illustrated in Figure 5, the original CDVAE [9] mainly consists of three parts: (1) a 3D encoder to encode the structure into the latent variable $z_{3D}$, (2) a property predictor to predict the lattice $\boldsymbol{L}$, the number of nodes in the unit cell $N$, and the proportion of each element in the composition $c$, (3) a 3D decoder to generate the structure from $z_{3D}, \boldsymbol{L}, N, c$ via the Score Matching with Langevin Dynamics (SMLD, Song and Ermon [49]) method. The training objective is composed of the loss functions on the three parts, *i.e.* the KL divergence between the encoded distribution and the standard normal distribution $\mathcal{L}_{KL}$, the aggregated prediction loss $\mathcal{L}_{AGG}$ and the denoising loss on the decoder $\mathcal{L}_{DEC}$. Formally, we have

$$\mathcal{L}_{ORI} = \mathcal{L}_{AGG} + \mathcal{L}_{DEC} + \beta D_{KL}\left(\mathcal{N}(\mu_{3D}, \sigma_{3D}^2 \boldsymbol{I})\|\mathcal{N}(0, \boldsymbol{I})\right).$$

We formulate $\mathcal{L}_{KL} = \beta D_{KL}(\mathcal{N}(\mu_{3D}, \sigma_{3D}^2\boldsymbol{I})\|\mathcal{N}(0, \boldsymbol{I}))$ for better comparison with the adapted method. $\beta$ is the hyper-parameter to balance the scale of the KL divergence and other loss functions.

To adapt the CDVAE framework to the CSP task, we apply two main changes. Firstly, for the encoder side, to take the composition as the condition, we apply an additional 1D prior encoder to encode the composition set into a latent distribution $\mathcal{N}(\mu_{1D}, \sigma_{1D}^2\boldsymbol{I})$ and minimize the KL divergence between the 3D and 1D distribution. The training objective is modified into

$$\mathcal{L}_{ADA} = \mathcal{L}_{AGG} + \mathcal{L}_{DEC} + \beta D_{KL}\left(\mathcal{N}(\mu_{3D}, \sigma_{3D}^2\boldsymbol{I})\|\mathcal{N}(\mu_{1D}, \sigma_{1D}^2\boldsymbol{I})\right).$$

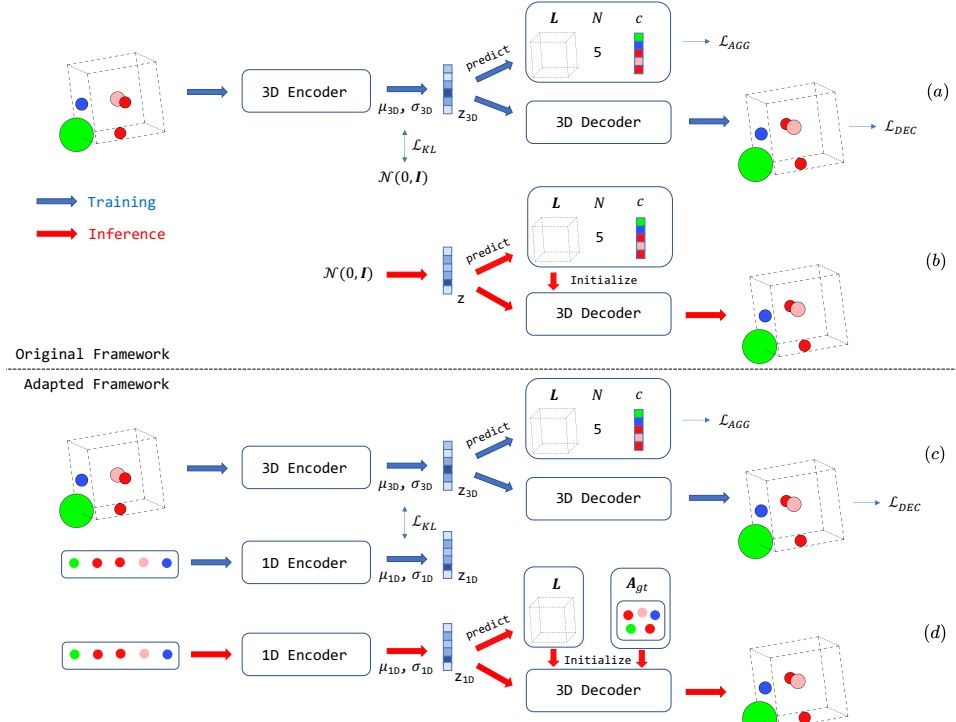

Figure 5: Overview of the original (a,b) and adapted (c,d) CDVAE. The key adaptations lie in two points. (1) We introduce an additional 1D prior encoder to fit the latent distribution of the given composition. (2) We initialize the generation procedure of the 3D decoder with the ground truth composition and keep the atom types unchanged to ensure the generated structure conforms to the given composition.

During the inference procedure, as the composition is given, the latent variable $z_{1D}$ is sampled from $\mathcal{N}(\mu_{1D}, \sigma_{1D}^2 \boldsymbol{I})$. For implementation, we apply a Transformer [51] without positional encoding as the 1D encoder to ensure the permutation invariance. Secondly, for the generation procedure, we apply the ground truth composition for initialization and keep the atom types unchanged during the Langevin dynamics to ensure the generated structure conforms to the given composition.

### B.3 Algorithms for Training and Sampling

Algorithm 1 summarizes the forward diffusion process as well as the training of the denoising model $\phi$, while Algorithm 2 illustrates the backward sampling process. They can maintain the symmetries if $\phi$ is delicately constructed. Notably, We apply the predictor-corrector sampler [50] to sample $\boldsymbol{F}_0$. In Algorithm 2, Line 7 refers to the predictor while Lines 9-10 correspond to the corrector.

---

**Algorithm 1** Training Procedure of DiffCSP

1: **Input:** lattice matrix $\boldsymbol{L}_0$, atom types $\boldsymbol{A}$, fractional coordinates $\boldsymbol{F}_0$, denoising model $\phi$, and the number of sampling steps $T$.
2: Sample $\boldsymbol{\epsilon_L} \sim \mathcal{N}(\boldsymbol{0}, \boldsymbol{I})$,$\boldsymbol{\epsilon_F} \sim \mathcal{N}(\boldsymbol{0}, \boldsymbol{I})$ and $t \sim \mathcal{U}(1, T)$.
3: $\boldsymbol{L}_t \leftarrow \sqrt{\bar{\alpha}_t}\boldsymbol{L}_0 + \sqrt{1 - \bar{\alpha}_t}\boldsymbol{\epsilon_L}$
4: $\boldsymbol{F}_t \leftarrow w(\boldsymbol{F}_0 + \sigma_t\boldsymbol{\epsilon_F})$
5: $\hat{\boldsymbol{\epsilon}}_L, \hat{\boldsymbol{\epsilon}}_F \leftarrow \phi(\boldsymbol{L}_t, \boldsymbol{F}_t, \boldsymbol{A}, t)$
6: $\mathcal{L}_L \leftarrow \|\boldsymbol{\epsilon_L} - \hat{\boldsymbol{\epsilon}}_L\|_2^2$
7: $\mathcal{L}_F \leftarrow \lambda_t\|\nabla_{\boldsymbol{F}_t} \log q(\boldsymbol{F}_t|\boldsymbol{F}_0) - \hat{\boldsymbol{\epsilon}}_F\|_2^2$
8: Minimize $\mathcal{L}_L + \mathcal{L}_F$

---

---

**Algorithm 2** Sampling Procedure of DiffCSP

---

1: **Input:** atom types $\boldsymbol{A}$, denoising model $\phi$, number of sampling steps $T$, step size of Langevin dynamics $\gamma$.
2: Sample $\boldsymbol{L}_T \sim \mathcal{N}(\boldsymbol{0}, \boldsymbol{I})$, $\boldsymbol{F}_T \sim \mathcal{U}(0, 1)$.
3: **for** $t \leftarrow T, \cdots, 1$ **do**
4:     Sample $\boldsymbol{\epsilon}_L, \boldsymbol{\epsilon}_F, \boldsymbol{\epsilon}'_F \sim \mathcal{N}(\boldsymbol{0}, \boldsymbol{I})$
5:     $\hat{\boldsymbol{\epsilon}}_L, \hat{\boldsymbol{\epsilon}}_F \leftarrow \phi(\boldsymbol{L}_t, \boldsymbol{F}_t, \boldsymbol{A}, t)$.
6:     $\boldsymbol{L}_{t-1} \leftarrow \frac{1}{\sqrt{\alpha_t}}(\boldsymbol{L}_t - \frac{\beta_t}{\sqrt{1-\bar{\alpha}_t}}\hat{\boldsymbol{\epsilon}}_L) + \sqrt{\beta_t \cdot \frac{1-\bar{\alpha}_{t-1}}{1-\bar{\alpha}_t}}\boldsymbol{\epsilon}_L$.
7:     $\boldsymbol{F}_{t-\frac{1}{2}} \leftarrow w(\boldsymbol{F}_t + (\sigma_t^2 - \sigma_{t-1}^2)\hat{\boldsymbol{\epsilon}}_F + \frac{\sigma_{t-1}\sqrt{\sigma_t^2-\sigma_{t-1}^2}}{\sigma_t}\boldsymbol{\epsilon}_F)$
8:     $\_, \hat{\boldsymbol{\epsilon}}_F \leftarrow \phi(\boldsymbol{L}_{t-1}, \boldsymbol{F}_{t-\frac{1}{2}}, \boldsymbol{A}, t-1)$.
9:     $d_t \leftarrow \gamma \sigma_{t-1}/\sigma_1$
10:     $\boldsymbol{F}_{t-1} \leftarrow w(\boldsymbol{F}_{t-\frac{1}{2}} + d_t\hat{\boldsymbol{\epsilon}}_F + \sqrt{2d_t}\boldsymbol{\epsilon}'_F)$.
11: **end for**
12: **Return** $\boldsymbol{L}_0, \boldsymbol{F}_0$.

---

### B.4 Hyper-parameters and Training Details

We acquire the origin datasets from CDVAE [9][4] and MPTS-52 [57][5]. We utilize the codebases from GN-OA [21][6], cG-SchNet [53][7] and CDVAE [9][8] for baseline implementations.

For the optimization methods, we apply the MEGNet [52] with 3 layers, 32 hidden states as property predictor. The model is trained for 1000 epochs with an Adam optimizer with learning rate $1 \times 10^{-3}$. As for the optimization algorithms, we apply RS, PSO, and BO according to Cheng et al. [21]. For RS and BO, We employ random search and TPE-based BO as implemented in Hyperopt [62][9]. Specifically, we choose observation quantile $\gamma$ as 0.25 and the number of initial random points as 200 for BO. For PSO, we used scikit-opt[10] and choose the momentum parameter $\omega$ as 0.8, the cognitive as 0.5, the social parameters as 0.5 and the size of population as 20.

For P-cG-SchNet, we apply the SchNet [29] with 9 layers, 128 hidden states as the backbone model. The model is trained for 500 epochs on each dataset with an Adam optimizer with initial learning rate $1 \times 10^{-4}$ and a Plateau scheduler with a decaying factor 0.5 and a patience of 10 epochs. We select the element proportion and the number of atoms in a unit cell as conditions for the CSP task. For CDVAE, we apply the DimeNet++ [63] with 4 layers, 256 hidden states as the encoder and the GemNet-T [64] with 3 layers, 128 hidden states as the decoder. We further apply a Transformer [51] model with 2 layers, 128 hidden states as the additional prior encoder as proposed in Appendix B.2. The model is trained for 3500, 4000, 1000, 1000 epochs for Perov-5, Carbon-24, MP-20 and MPTS-52 respectively with an Adam optimizer with initial learning rate $1 \times 10^{-3}$ and a Plateau scheduler with a decaying factor 0.6 and a patience of 30 epochs. For our DiffCSP, we utilize the setting of 4 layer, 256 hidden states for Perov-5 and 6 layer, 512 hidden states for other datasets. The dimension of the Fourier embedding is set to $k = 256$. We apply the cosine scheduler with $s = 0.008$ to control the variance of the DDPM process on $\boldsymbol{L}_t$, and an exponential scheduler with $\sigma_1 = 0.005, \sigma_T = 0.5$ to control the noise scale of the score matching process on $\boldsymbol{F}_t$. The diffusion step is set to $T = 1000$. Our model is trained for 3500, 4000, 1000, 1000 epochs for Perov-5, Carbon-24, MP-20 and MPTS-52 with the same optimizer and learning rate scheduler as CDVAE. For the step size $\gamma$ in Langevin dynamics for the structure prediction task, we apply $\gamma = 5 \times 10^{-7}$ for Perov-5, $1 \times 10^{-5}$ for MP-20 and MPTS-52, and for Carbon-24, we apply $\gamma = 5 \times 10^{-6}$ to predict one sample and $\gamma = 5 \times 10^{-7}$ for multiple samples. For the ab initio generation and optimization task on Perov-5, Carbon-24 and MP-20, we apply $\gamma = 1 \times 10^{-6}, 1 \times 10^{-5}, 5 \times 10^{-6}$, respectively. All models are trained on GeForce RTX 3090 GPU.

---

[4] https://github.com/txie-93/cdvae/tree/main/data
[5] https://github.com/sparks-baird/mp-time-split
[6] http://www.comates.group/links?software=gn_oa
[7] https://github.com/atomistic-machine-learning/cG-SchNet
[8] https://github.com/txie-93/cdvae
[9] https://github.com/hyperopt/hyperopt
[10] https://github.com/guofei9987/scikit-opt

# C  Exploring the Effects of Sampling and Candidate Ranking

## C.1  Impact of Sampling Numbers

Figure 6 illustrates the impact of sampling numbers on the match rate. The match rate of all methods increases when sampling more candidates, and DiffCSP outperforms the baselines methods under the arbitrary number of samples.

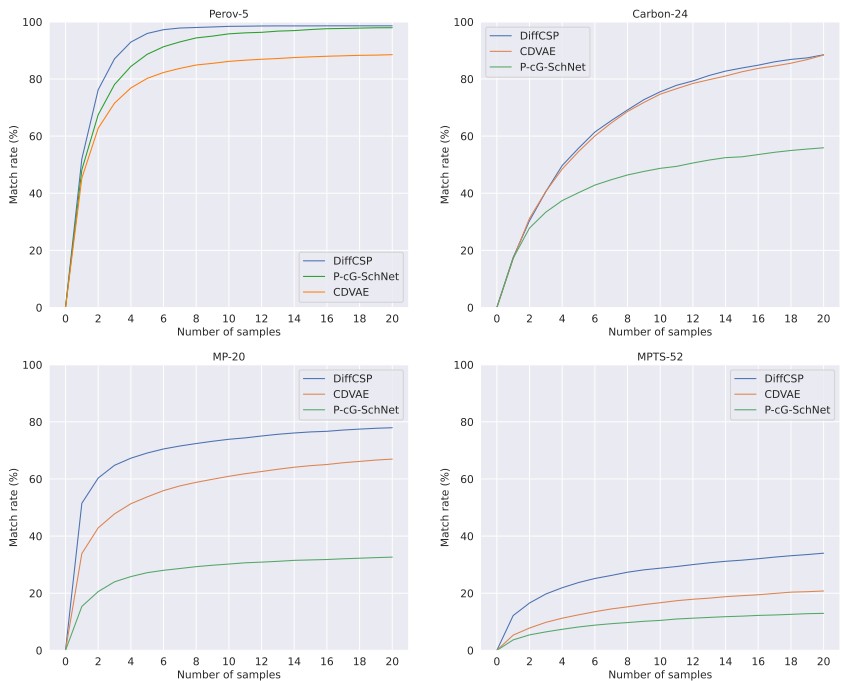

Figure 6: Comparison on different number of samples.

## C.2  Diversity

We further evaluate the diversity by yielding the CrystalNN [65] fingerprint of each generated structure, calculating the L2-distances among all pairs in the 20 samples of each composition, collating the mean and max value of the distances, and finally averaging the results from all testing candidates. We list the diversity of CDVAE and DiffCSP on Perov-5 and MP-20 in Table 5.

Table 5: Comparison on the diversity.

|  | Perov-5 | | MP-20 | |
|---|---|---|---|---|
|  | Mean | Max | Mean | Max |
| CDVAE | 0.3249 | 0.7316 | **0.3129** | **0.6979** |
| DiffCSP | **0.3860** | **0.8911** | 0.2030 | 0.5292 |

## C.3  Ranking among Multiple Candidates

In § 5.1, we match each candidate with the ground truth structure to pick up the best sample. However, in real CSP scenarios, the ground truth structure is not available, necessitating a confidence model to rank the generated candidates. To address this, we develop three types of confidence models to score each sample for ranking from different perspectives: **Energy Predictor (EP).** Since lower formation energy typically leads to more stable structures, we directly train a predictor using energy labels and apply the negative of the predicted

Table 6: Results on different confidence models. *Oracle* means applying the negative RMSD against the ground truth as the ranking score.

|  | Match rate (%) ↑ | RMSE ↓ |
|---|---|---|
| EP | 51.96 | 0.0589 |
| MD (d=0.1) | 60.30 | 0.0357 |
| MD (d=0.3) | 60.13 | 0.0382 |
| MD (d=0.5) | 59.20 | 0.0469 |
| CS | 58.81 | 0.0443 |
| Oracle | 77.93 | 0.0492 |

energy as the confidence score. **Match Discriminator (MD).** Inspired by Diffdock [12], we first generate five samples for each composition in the training/validation set using DiffCSP and calculate their RMSDs with the ground truth. We then train a binary classifier to predict whether the sample matches the ground truth with an RMSD below a threshold $d$. The predicted probability serves as the score. **Contrastive Scorer (CS).** Drawing inspiration from CLIP [66], we train a contrastive model between a 1D and 3D model to align the corresponding compositions and structures. The inner product of the 1D and 3D models is used as the score. We select the Top-1 result among the 20 candidates ranked by each confidence model on the MP-20 dataset, as shown in Table 6. The results indicate that MD and CS perform relatively better, but there remains a gap between the heuristic ranking models and the oracle ranker. Designing powerful ranking models is an essential problem, which we leave for future research.

## D   Impact of Noise Schedulers

We explore the noise schedulers from three perspectives and list the results in Table 7 and 8. **1.** For lattices, we originally use the cosine scheduler with $= 0.008$, and we change it into the linear and sigmoid schedulers with $\beta_1 = 0.0001, \beta_T = 0.02$. We find that the linear scheduler yields comparable results, while the sigmoid scheduler hinders the performance. **2.** For fractional coordinates, we use the exponential scheduler with $\sigma_{min} = 0.005$, $\sigma_{max} = 0.5$, and we change the value of $\sigma_{max}$ into 0.1 and 1.0. Results show that the small-$\sigma_{max}$ variant performs obviously worse, as only sufficiently large $\sigma_{max}$ could approximate the prior uniform distribution. We visualize the PDF curves in Figure 7 for better understanding. **3.** For atom types, we conduct similar experiments as lattices, and the results indicate that the original cosine scheduler performs better. In conclusion, we suggest applying the proposed noise schedulers.

Table 7: CSP results on different schedulers. DiffCSP utilizes cosine scheduler on $L$ and $\sigma_{max} = 0.5$ on $F$.

|  | Match rate (%) ↑ | RMSE ↓ |
|---|---|---|
| DiffCSP | 51.49 | 0.0631 |
| *Schedulers of $L$* | | |
| Linear | 50.06 | 0.0590 |
| Sigmoid | 45.24 | 0.0664 |
| *Schedulers of $F$* | | |
| $\sigma_{max} = 0.1$ | 32.56 | 0.0913 |
| $\sigma_{max} = 1.0$ | 47.89 | 0.0675 |

Table 8: Ab initio generation results on different type schedulers. DiffCSP utilizes cosine scheduler on $A$.

|  | Validity | | Coverage | |
|---|---|---|---|---|
|  | Struct.(%) | Comp.(%) | Recall | Precision |
| DiffCSP | 100.00 | 83.25 | 99.71 | 99.76 |
| Linear | 99.70 | 79.78 | 98.29 | 99.48 |
| Sigmoid | 99.88 | 81.59 | 99.33 | 99.55 |

## E   Learning Curves of Different Variants

We plot the curves of training and validation loss of different variants proposed in § 5.3 in Figure 8 with the following observations. **1.** The multi-graph methods struggle with higher training and validation loss, as the edges constructed under different disturbed lattices vary significantly, complicating the training procedure. **2.** The Fourier transformation, expanding the relative coordinates and maintaining the periodic translation invariance, helps the model converge faster at the beginning of the training procedure. **3.** The variant utilizing the fully connected graph without the Fourier transformation (named "DiffCSP w/o FT" in Figure 8) encounters obvious overfitting as the periodic translation invariance is violated, highlighting the necessity to leverage the desired invariance into the model.

## F   Comparison with DFT-based Methods

### F.1   Implementation Details

We utilize USPEX [59], a DFT-based software equipped with the evolutionary algorithm to search for stable structures. We use 20 populations in each generation, and end up with 20 generations for each compound. We set 60% of the lowest-enthalpy structures allowed to produce the next generation through heredity (50%), lattice mutation (10%), and atomic permutation (20%). Moreover, two

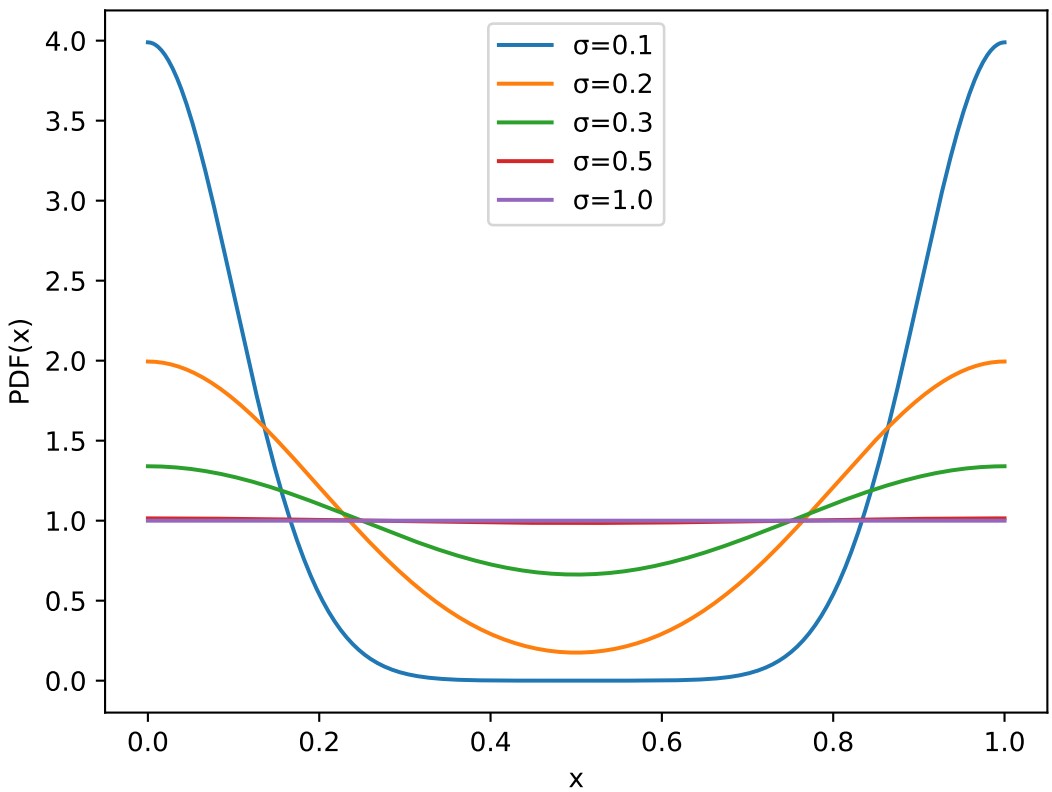

Figure 7: PDF curves of the wrapped normal distribution with periodic as 1 and different noise scales. It can be find that $\sigma = 0.5$ is practically large enough to approximate the prior uniform distribution.

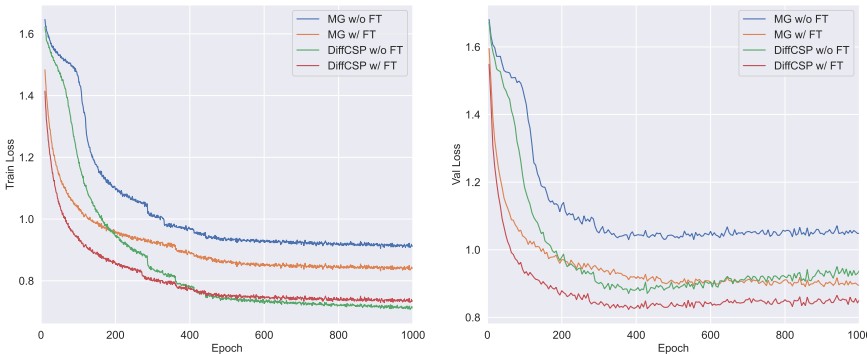

Figure 8: Learning curves of different variants proposed in § 5.3. MG and FT denote multi-graph edge construction and Fourier transformation, respectively.

lowest-enthalpy structures are allowed to survive into the next generation. The structural relaxations are calculated by the frozen-core all-electron projector augmented wave (PAW) method [67] as implemented in the Vienna ab initio simulation package (VASP) [68]. Each sample's calculation is performed using one node with 48 cores (Intel(R) Xeon(R) CPU E5-2692 v2 @ 2.20GHz), while the populations within the same generation are concurrently computed across 20 nodes in parallel. The exchange-correlation energy is treated within the generalized gradient approximation (GGA), using the Perdew-Burke-Ernzerhof (PBE) function [69].

### F.2 Results

We select 10 binary and 5 ternary compounds in MP-20 testing set and compare our model with USPEX. For our method, we sample 20 candidates for each compound following the setting in Table 1. For USPEX, we apply 20 generations, 20 populations for each compound, and select the best sample in each generation, leading to 20 candidates as well. We list the minimum RMSD of each compound in Table 9, and additionally summarize the match rate over the 15 compounds, the averaged RMSD over the matched structures, and the averaged inference time to generate 20 candidates for each compound in Table 10. The results show that DiffCSP correctly predicts more structures with higher match rate and significantly lower time cost.

Table 9: The minimum RMSD of 20 candidates of USPEX and DiffCSP, "N/A" means none of the candidates match with the ground truth.

| Binary | $Co_2Sb_2$ | $Sr_2O_4$ | $AlAg_4$ | $YMg_3$ | $Cr_4Si_4$ |
|---|---|---|---|---|---|
| USPEX | 0.0008 | 0.0121 | N/A | 0.0000 | N/A |
| DiffCSP | 0.0005 | 0.0133 | 0.0229 | 0.0003 | 0.0066 |

| Binary | $Sn_4Pd_4$ | $Ag_6O_2$ | $Co_4B_2$ | $Ba_2Cd_6$ | $Bi_2F_8$ |
|---|---|---|---|---|---|
| USPEX | 0.0184 | 0.0079 | 0.0052 | N/A | N/A |
| DiffCSP | 0.0264 | N/A | N/A | 0.0028 | N/A |

| Ternary | $KZnF_3$ | $Cr_3CuO_8$ | $Bi_4S_4Cl_4$ | $Si_2(CN_2)_4$ | $Hg_2S_2O_8$ |
|---|---|---|---|---|---|
| USPEX | 0.0123 | N/A | N/A | N/A | 0.0705 |
| DiffCSP | 0.0006 | 0.0482 | 0.0203 | N/A | 0.0473 |

Table 10: Overall results over the 15 selected compounds.

| | Match Rate (%)↑ | Avg. RMSD↓ | Avg. Time↓ |
|---|---|---|---|
| USPEX | 53.33 | **0.0159** | 12.5h |
| DiffCSP | **73.33** | 0.0172 | **10s** |

## G  Extension to More General Tasks

### G.1  Overview

Our method mainly focuses on CSP, aiming at predicting the stable structures from the fixed composition $A$. We first enable the generation on atom types and extend to ab initio generation task in § G.2, and then adopt the energy guidance for property optimization in § G.3. Figure 9 illustrated the differences and connections of CSP, ab initio generation, and property optimization. Besides, CDVAE [9] proposes a reconstruction task specific for VAE-based models, which first encodes the ground truth structure, and require the model to recover the input structure. As our method follows a diffusion-based framework instead of VAE, it is unsuitable to conduct the reconstruction task of our method.

### G.2  Ab Initio Generation

Apart from $L$ and $F$, the ab initio generation task additionally requires the generation on $A$. As the atom types can be viewed as either $N$ discrete variables in $h$ classes, or the one-hot representation $A \in \mathbb{R}^{h \times N}$ in continuous space. We apply two lines of diffusion processes for the type generation, which are detailedly shown as follows.

**Multinomial Diffusion for Discrete Generation** Regarding the atom types as discrete features, we apply the multinomial diffusion with the following forward diffusion process [70, 42, 71],

$$q(\boldsymbol{A}_t|\boldsymbol{A}_0) = \mathcal{C}(\boldsymbol{A}_t|\bar{\alpha}_t \boldsymbol{A}_0 + \frac{(1 - \bar{\alpha}_t)}{h}\mathbf{1}_h\mathbf{1}_N^\top), \tag{18}$$

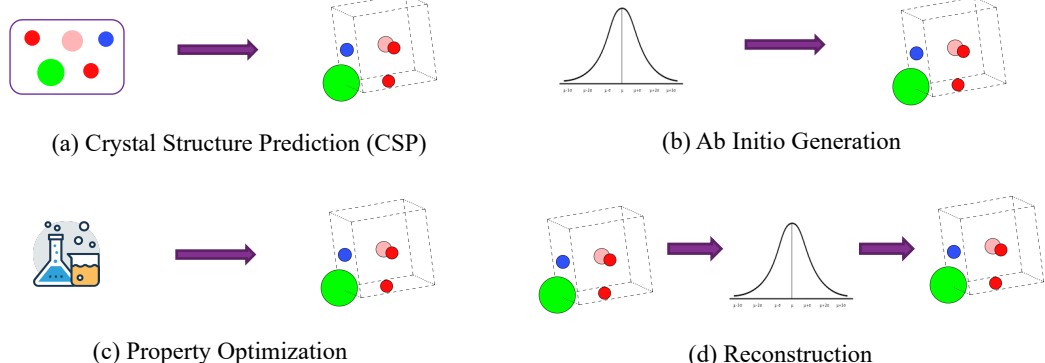

(a) Crystal Structure Prediction (CSP)

(b) Ab Initio Generation

(c) Property Optimization

(d) Reconstruction

Figure 9: Different tasks for crystal generation. Our approach mainly focuses on the CSP task (a), and is capable to extend into the ab initio generation task (b) by further generating the composition, and the property optimization task (c) via introducing the guidance on the target property. The reconstruction task (d) in Xie et al. [9] is specific for VAE, which is unnecessary for our diffusion-based method.

where $\mathbf{1}_h \in \mathbb{R}^{h \times 1}, \mathbf{1}_N \in \mathbb{R}^{N \times 1}$ are vectors with all elements setting to one, $\boldsymbol{A}_0$ is the one-hot representation of the origin composition, and the function $\mathcal{C}(\cdot)$ samples the multinomial distribution with the conditional probability and returns the one-hot representation of $\boldsymbol{A}_t$.

The corresponding backward generation process is defined as

$$p(\boldsymbol{A}_{t-1}|\mathcal{M}_t) = \mathcal{C}(\boldsymbol{A}_{t-1}|\tilde{\theta}_t / \sum_{k=1}^{h} \tilde{\theta}_{t,k}), \tag{19}$$

where

$$\tilde{\theta}_t = \left( \alpha_t \boldsymbol{A}_t + \frac{(1-\alpha_t)}{h} \mathbf{1}_h \mathbf{1}_N^\top \right) \odot \left( \bar{\alpha}_t \hat{\epsilon}_{\boldsymbol{A}}(\mathcal{M}_t, t) + \frac{(1-\bar{\alpha}_t)}{h} \mathbf{1}_h \mathbf{1}_N^\top \right), \tag{20}$$

and $\hat{\epsilon}_{\boldsymbol{A}} \in \mathbb{R}^{h \times N}$ is predicted by the denoising model. We further find that specific for $t = 1$, $\boldsymbol{A}_0 = \mathrm{argmax}(\hat{\epsilon}_{\boldsymbol{A}}(\mathcal{M}_1, 1))$ works better.

The training objective for multinomial diffusion is

$$\mathcal{L}_{\boldsymbol{A},\text{discrete}} = \mathbb{E}_{\boldsymbol{A}_t \sim q(\boldsymbol{A}_t|\boldsymbol{A}_0), t \sim \mathcal{U}(1,T)} \Big[ \mathrm{KL}(q(\boldsymbol{A}_{t-1}|\boldsymbol{A}_t) || p(\boldsymbol{A}_{t-1}|\boldsymbol{A}_t)) \Big]. \tag{21}$$

**One-hot Diffusion for Continuous Generation** Another approach is to simply consider the composition $\boldsymbol{A}$ as a continuous variable in real space $\mathbb{R}^{h \times N}$, which enables the application of standard DDPM-based method [41]. Similar to Eq. (2)-(4), the forward diffusion process is defined as

$$q(\boldsymbol{A}_t|\boldsymbol{A}_0) = \mathcal{N}\Big( \boldsymbol{L}_t | \sqrt{\bar{\alpha}_t} \boldsymbol{A}_0, (1-\bar{\alpha}_t)\boldsymbol{I} \Big). \tag{22}$$

And the backward generation process is defined as

$$p(\boldsymbol{A}_{t-1}|\mathcal{M}_t) = \mathcal{N}(\boldsymbol{A}_{t-1}|\mu_{\boldsymbol{A}}(\mathcal{M}_t), \sigma_{\boldsymbol{A}}^2(\mathcal{M}_t)\boldsymbol{I}), \tag{23}$$

where $\mu_{\boldsymbol{A}}(\mathcal{M}_t) = \frac{1}{\sqrt{\alpha_t}} \Big( \boldsymbol{A}_t - \frac{\beta_t}{\sqrt{1-\bar{\alpha}_t}} \hat{\epsilon}_{\boldsymbol{A}}(\mathcal{M}_t, t) \Big), \sigma_{\boldsymbol{A}}^2(\mathcal{M}_t) = \beta_t \frac{1-\bar{\alpha}_{t-1}}{1-\bar{\alpha}_t}$. The denoising term $\hat{\epsilon}_{\boldsymbol{A}}(\mathcal{M}_t, t) \in \mathbb{R}^{h \times N}$ is predicted by the model.

The training objective for one-hot diffusion is

$$\mathcal{L}_{\boldsymbol{A},\text{continuous}} = \mathbb{E}_{\epsilon_{\boldsymbol{A}} \sim \mathcal{N}(0,\boldsymbol{I}), t \sim \mathcal{U}(1,T)} [\|\epsilon_{\boldsymbol{A}} - \hat{\epsilon}_{\boldsymbol{A}}(\mathcal{M}_t, t)\|_2^2]. \tag{24}$$

The entire objective for training the joint diffusion model on $\boldsymbol{L}, \boldsymbol{F}, \boldsymbol{A}$ is combined as

$$\mathcal{L}_{\mathcal{M}} = \lambda_{\boldsymbol{L}} \mathcal{L}_{\boldsymbol{L}} + \lambda_{\boldsymbol{F}} \mathcal{L}_{\boldsymbol{F}} + \lambda_{\boldsymbol{A}} \mathcal{L}_{\boldsymbol{A}}. \tag{25}$$

We select $\lambda_L = \lambda_F = 1, \lambda_A = 0$ for the CSP task as $A$ is fixed during the generation process, and $\lambda_L = \lambda_F = 1, \lambda_A = 20$ for the ab initio generation task to balance the scale of each loss component. Specifically, we do not optimize $\mathcal{L}_A$ on the Carbon-24 dataset, as all atoms in this dataset are carbon.

**Sample Structures with Arbitrary Numbers of Atoms** As the number of atoms in a unit cell (*i.e.* $N$) is unchanged during the generation process, we first sample $N$ according to the distribution of $N$ in the training set, which is similar to Hoogeboom et al. [41]. The sampled distribution $p(\mathcal{M})$ can be more concisely described as $p(\mathcal{M}, N) = p(N)p(\mathcal{M}|N)$. The former term $p(N)$ is sampled from pre-computed data distribution, and the latter conditional distribution $p(\mathcal{M}|N)$ is modeled by DiffCSP.

**Evaulation Metrics** The results are evaluated from three perspectives. **Validity**: We consider both the structural validity and the compositional validity. The structural valid rate is calculated as the percentage of the generated structures with all pairwise distances larger than 0.5Å, and the generated composition is valid if the entire charge is neutral as determined by SMACT [72]. **Coverage**: It measures the structural and compositional similarity between the testing set $\mathcal{S}_t$ and the generated samples $\mathcal{S}_g$. Specifically, letting $d_S(\mathcal{M}_1, \mathcal{M}_2), d_C(\mathcal{M}_1, \mathcal{M}_2)$ denote the $L2$ distances of the CrystalNN structural fingerprints [65] and the normalized Magpie compositional fingerprints [73], the COVerage Recall (COV-R) is determined as COV-R $= \frac{1}{|\mathcal{S}_t|}|\{\mathcal{M}_i|\mathcal{M}_i \in \mathcal{S}_t, \exists \mathcal{M}_j \in \mathcal{S}_g, d_S(\mathcal{M}_i, \mathcal{M}_j) < \delta_S, d_C(\mathcal{M}_i, \mathcal{M}_j) < \delta_C\}|$ where $\delta_S, \delta_C$ are pre-defined thresholds. The COVerage Precision (COV-P) is defined similarly by swapping $\mathcal{S}_g, \mathcal{S}_t$. **Property statistics**: We calculate three kinds of Wasserstein distances between the generated and testing structures, in terms of density, formation energy, and the number of elements [9], denoted as $d_\rho, d_E$ and $d_{\text{elem}}$, individually. The validity and coverage metrics are calculated on 10,000 generated samples, and the property metrics are evaluated on a subset with 1,000 samples passing the validity test.

**Results** We denote the abovementioned discrete and continuous generation methods as DiffCSP-D, DiffCSP-C, respectively. The results of the two variants and the strongest baseline CDVAE on MP-20 are provided in Table 11. We observe that DiffCSP-D yields slightly lower validity and coverage rates than DiffCSP-C. Moreover, DiffCSP-D tends to generate structures with more types of elements, which is far from the data distribution. Hence, we select DiffCSP-C for the other experiments in Table 4 (abbreviated as DiffCSP). We further find that DiffCSP-C supports the property optimization task in the next section. Besides, both variants have lower composition validity than CDVAE, implying that more powerful methods for composition generation are required. As our paper mainly focuses on the CSP task, we leave this for future studies.

### G.3 Property Optimization

On top of DiffCSP-C, we further equip our method with energy guidance [74, 75] for property optimization. Specifically we train a time-dependent property predictor $E(L_t, F_t, A_t, t)$ with the same message passing blocks as Eq. (6)-(8). And the final prediction is acquired by the final layer as

$$E = \varphi_E\left(\frac{1}{N}\sum_{i=1}^{N} h_i^{(S)}\right). \tag{26}$$

And the gradients *w.r.t.* $L, F, A$ additionally guide the generation process. As the inner product term $L^\top L$ is O(3) invariant, and the Fourier transformation term $\phi_{\text{FT}}(f_j - f_i)$ is periodic translation invariant, the predicted energy $E$ is periodic E(3) invariant. That is,

$$E(QL_t, w(F_t + t\mathbf{1}^\top, A_t, t) = E(L_t, F_t, A_t, t). \tag{27}$$

Table 11: Results on MP-20 ab initio generation task.

| Data | Method | Validity (%) ↑ | | Coverage (%) ↑ | | Property ↓ | | |
| | | Struc. | Comp. | COV-R | COV-P | $d_\rho$ | $d_E$ | $d_{\text{elem}}$ |
|---|---|---|---|---|---|---|---|---|
| MP-20 | CDVAE [9] | **100.0** | **86.70** | 99.15 | 99.49 | 0.6875 | 0.2778 | 1.432 |
| | DiffCSP-D | 99.70 | 83.11 | 99.68 | 99.53 | **0.1730** | 0.1366 | 0.9959 |
| | DiffCSP-C | **100.0** | 83.25 | **99.71** | **99.76** | 0.3502 | **0.1247** | **0.3398** |

**Algorithm 3** Energy-Guided Sampling Procedure of DiffCSP(-C)

---

1: **Input:** denoising model $\phi$, energy predictor $E$, step size of Langevin dynamics $\gamma$, guidance magnitude $s$, input structure before optimization $\mathcal{M} = (\boldsymbol{L}, \boldsymbol{F}, \boldsymbol{A})$, number of sampling steps $T'$, maximum number of sampling steps $T$.
2: **if** T'=T **then**
3:     Sample $\boldsymbol{L}_{T'} \sim \mathcal{N}(\boldsymbol{0}, \boldsymbol{I}), \boldsymbol{F}_{T'} \sim \mathcal{U}(0, 1), \boldsymbol{A}_{T'} \sim \mathcal{N}(\boldsymbol{0}, \boldsymbol{I})$.
4: **else**
5:     Sample $\boldsymbol{L}_{T'} \sim q(\boldsymbol{L}_{T'}|\boldsymbol{L}), \boldsymbol{F}_{T'} \sim q(\boldsymbol{F}_{T'}|\boldsymbol{F}), \boldsymbol{A}_{T'} \sim q(\boldsymbol{A}_{T'}|\boldsymbol{A})$ according to Eq. (2),(5) and (22).
6: **end if**
7: **for** $t \leftarrow T', \cdots, 1$ **do**
8:     Sample $\boldsymbol{\epsilon}_L, \boldsymbol{\epsilon}_F, \boldsymbol{\epsilon}_A, \boldsymbol{\epsilon}'_F \sim \mathcal{N}(\boldsymbol{0}, \boldsymbol{I})$
9:     $\hat{\boldsymbol{\epsilon}}_L, \hat{\boldsymbol{\epsilon}}_F, \hat{\boldsymbol{\epsilon}}_A \leftarrow \phi(\boldsymbol{L}_t, \boldsymbol{F}_t, \boldsymbol{A}_t, t)$.
10:     Acquire $\nabla_L E, \nabla_F E, \nabla_A E$ from $E(\boldsymbol{L}_t, \boldsymbol{F}_t, \boldsymbol{A}_t, t)$.
11:     $\boldsymbol{L}_{t-1} \leftarrow \frac{1}{\sqrt{\alpha_t}}(\boldsymbol{L}_t - \frac{\beta_t}{\sqrt{1-\bar{\alpha}_t}}\hat{\boldsymbol{\epsilon}}_L) - s\beta_t \cdot \frac{1-\bar{\alpha}_{t-1}}{1-\bar{\alpha}_t}\nabla_L E + \sqrt{\beta_t \cdot \frac{1-\bar{\alpha}_{t-1}}{1-\bar{\alpha}_t}}\boldsymbol{\epsilon}_L$.
12:     $\boldsymbol{A}_{t-1} \leftarrow \frac{1}{\sqrt{\alpha_t}}(\boldsymbol{A}_t - \frac{\beta_t}{\sqrt{1-\bar{\alpha}_t}}\hat{\boldsymbol{\epsilon}}_A) - s\beta_t \cdot \frac{1-\bar{\alpha}_{t-1}}{1-\bar{\alpha}_t}\nabla_A E + \sqrt{\beta_t \cdot \frac{1-\bar{\alpha}_{t-1}}{1-\bar{\alpha}_t}}\boldsymbol{\epsilon}_A$.
13:     $\boldsymbol{F}_{t-\frac{1}{2}} \leftarrow w(\boldsymbol{F}_t + (\sigma_t^2 - \sigma_{t-1}^2)\hat{\boldsymbol{\epsilon}}_F - s\frac{\sigma_{t-1}^2(\sigma_t^2 - \sigma_{t-1}^2)}{\sigma_t^2}\nabla_F E + \frac{\sigma_{t-1}\sqrt{\sigma_t^2 - \sigma_{t-1}^2}}{\sigma_t}\boldsymbol{\epsilon}_F)$
14:     $\_, \hat{\boldsymbol{\epsilon}}_F, \_ \leftarrow \phi(\boldsymbol{L}_{t-1}, \boldsymbol{F}_{t-\frac{1}{2}}, \boldsymbol{A}_{t-1}, t-1)$.
15:     $d_t \leftarrow \gamma\sigma_{t-1}/\sigma_1$
16:     $\boldsymbol{F}_{t-1} \leftarrow w(\boldsymbol{F}_{t-\frac{1}{2}} + d_t\hat{\boldsymbol{\epsilon}}_F + \sqrt{2d_t}\boldsymbol{\epsilon}'_F)$.
17: **end for**
18: **Return** $\boldsymbol{L}_0, \boldsymbol{F}_0, \text{argmax}(\boldsymbol{A}_0)$.

---

Table 12: Results on property optimization task. The results of baselines are from Xie et al. [9].

| Method | Perov-5 | | | Carbon-24 | | | MP-20 | | |
|---|---|---|---|---|---|---|---|---|---|
| | SR5 | SR10 | SR15 | SR5 | SR10 | SR15 | SR5 | SR10 | SR15 |
| FTCP | 0.06 | 0.11 | 0.16 | 0.00 | 0.00 | 0.00 | 0.02 | 0.04 | 0.05 |
| Cond-DFC-VAE | 0.55 | 0.64 | 0.69 | – | – | – | – | – | – |
| CDVAE | 0.52 | 0.65 | 0.79 | 0.00 | 0.06 | 0.06 | 0.78 | 0.86 | 0.90 |
| DiffCSP | **1.00** | **1.00** | **1.00** | **0.50** | **0.69** | **0.69** | **0.82** | **0.98** | **1.00** |

Taking gradient to both sides *w.r.t.* $\boldsymbol{L}, \boldsymbol{F}, \boldsymbol{A}$, respectively, we have

$$\boldsymbol{Q}^\top \nabla_{\boldsymbol{L}'_t} E(\boldsymbol{L}'_t, w(\boldsymbol{F}_t + \boldsymbol{t}\boldsymbol{1}^\top), \boldsymbol{A}_t, t)|_{\boldsymbol{L}'_t = \boldsymbol{Q}\boldsymbol{L}_t} = \nabla_{\boldsymbol{L}_t} E(\boldsymbol{L}_t, \boldsymbol{F}_t, \boldsymbol{A}_t, t), \tag{28}$$

$$\nabla_{\boldsymbol{F}'_t} E(\boldsymbol{Q}\boldsymbol{L}_t, \boldsymbol{F}'_t, \boldsymbol{A}_t, t)|_{\boldsymbol{F}'_t = w(\boldsymbol{F}_t + \boldsymbol{t}\boldsymbol{1}^\top)} = \nabla_{\boldsymbol{F}_t} E(\boldsymbol{L}_t, \boldsymbol{F}_t, \boldsymbol{A}_t, t), \tag{29}$$

$$\nabla_{\boldsymbol{A}_t} E(\boldsymbol{Q}\boldsymbol{L}_t, w(\boldsymbol{F}_t + \boldsymbol{t}\boldsymbol{1}^\top), \boldsymbol{A}_t, t) = \nabla_{\boldsymbol{A}_t} E(\boldsymbol{L}_t, \boldsymbol{F}_t, \boldsymbol{A}_t, t), \tag{30}$$

which implies that the gradient to $\boldsymbol{L}_t$ is O(3) equivariant, and the gradient to $\boldsymbol{F}_t$ and $\boldsymbol{A}_t$ is periodic E(3) invariant. Such symmetries maintain that the introduction of energy guidance does not violate the periodic E(3) invariance of the marginal distribution.

The detailed algorithm for energy-guided sampling is provided in Algorithm 3. We find that $s = 50$ works well on the three datasets. We evaluate the performance of the energy-guided model with the same metrics as Xie et al. [9]. We sample 100 structures from the testing set for optimization. For each structure, we apply $T = 1,000$ and $T' = 100, 200, \cdots, 1,000$, leading to 10 optimized structures. We use the same independent property predictor as in Xie et al. [9] to select the best one from the 10 structures. We calculate the success rate (**SR**) as the percentage of the 100 optimized structures achieving 5, 10, and 15 percentiles of the property distribution. We select the formation energy per atom as the target property, and provide the results on Perov-5, Carbon-24 and MP-20 in Table 12, showcasing the notable superiority of DiffCSP over the baseline methods.

Aside from the Carbon-24 dataset, the composition is flexible in the above experiments. We also attempt to follow the CSP setting and optimize the crystal structures for lower energy based on the fixed composition. We visualize eight cases in Figure 10 and calculate the energies of the structures before and after optimization by VASP [68]. Results show that our method is capable to search novel structures with lower energies compared to existing ones.

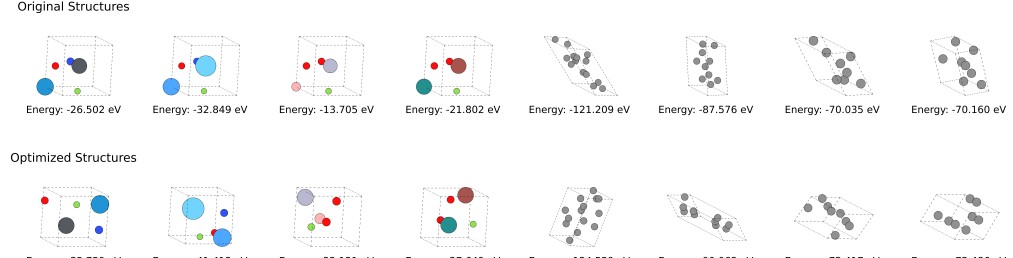

Figure 10: Visualization of 8 pairs of structures before and after optimization.

Table 13: GPU hours for yielding 20 candidates over the testing set.

|  | Perov-5 | Carbon-24 | MP-20 | MPTS-52 |
|---|---|---|---|---|
| P-cG-SchNet | 2.1 | 3.0 | 10.0 | 22.5 |
| CDVAE | 21.9 | 9.8 | 95.2 | 178.0 |
| DiffCSP | 1.5 | 3.2 | 18.4 | 34.0 |

Table 14: Results on MP-20 with different inference steps.

|  | Steps | Match rate (%)↑ | RMSD↓ |
|---|---|---|---|
| DiffCSP | 1,000 | 51.49 | 0.0631 |
|  | 5,000 | 52.95 | 0.0541 |
| CDVAE | 1,000 | 30.71 | 0.1288 |
|  | 5,000 | 33.90 | 0.1045 |

## H Computational Cost for Inference

We provide the GPU hours (GeForce RTX 3090) for different generative methods to predict 20 candidates on the 4 datasets. Table 13 demonstrates the diffusion-based models (CDVAE and our DiffCSP) are slower than P-cG-SchNet. Yet, the computation overhead of our method is still acceptable given its clearly better performance than P-cG-SchNet. Additionally, our DiffCSP is much faster than CDVAE across all datasets mainly due to the fewer generation steps. CDVAE requires 5,000 steps for each generation, whereas our approach only requires 1,000 steps. We further compare the performance of CDVAE and DiffCSP with 1,000 and 5,000 generation steps on MP-20 in Table 14. Our findings indicate that both models exhibit improved performance with an increased number of steps. Notably, DiffCSP with 1,000 steps outperforms CDVAE with 5,000 steps.

## I Error Bars

We provide a single run to generate 20 candidates in Table 1. We apply two more inferences of each generative method on Perov-5 and MP-20. Table 15 shows similar results as Table 1.

Table 15: Results on Perov-5 and MP-20 with error bars.

|  | # of samples | Perov-5 | | MP-20 | |
|---|---|---|---|---|---|
|  |  | Match rate (%)↑ | RMSE↓ | Match rate (%)↑ | RMSE↓ |
| P-cG-Schnet [53] | 1 | 47.34±0.63 | 0.4170±0.0006 | 15.59±0.41 | 0.3747±0.0020 |
|  | 20 | 97.92±0.02 | 0.3464±0.0004 | 32.70±0.12 | 0.3020±0.0002 |
| CDVAE [9] | 1 | 45.31±0.49 | 0.1123±0.0026 | 33.93±0.15 | 0.1069±0.0018 |
|  | 20 | 88.20±0.26 | 0.0473±0.0007 | 67.20±0.23 | 0.1012±0.0016 |
| DiffCSP | 1 | 52.35±0.26 | 0.0778±0.0030 | 51.89±0.30 | 0.0611±0.0015 |
|  | 20 | **98.58±0.02** | **0.0129±0.0003** | **77.85±0.23** | **0.0493±0.0011** |

## J More Visualizations

In this section, we first present additional visualizations of the predicted structures from DiffCSP and other generative methods in Figure 11. In line with Figure 3, our DiffCSP provides more accurate predictions compared with the baseline methods. Figure 12 illustrates 16 generated structures on Perov-5, Carbon-24 and MP-20. The visualization shows the capability of DiffCSP to generate diverse

structures. We further visualize the generation process of 5 structures from MP-20 in Figure 13. We find that the generated structure $\mathcal{M}_0$ is periodically translated from the ground truth structure, indicating that the marginal distribution $p(\mathcal{M}_0)$ follows the desired periodic translation invariance. We provide the detailed generation process in the Supplementary Materials.

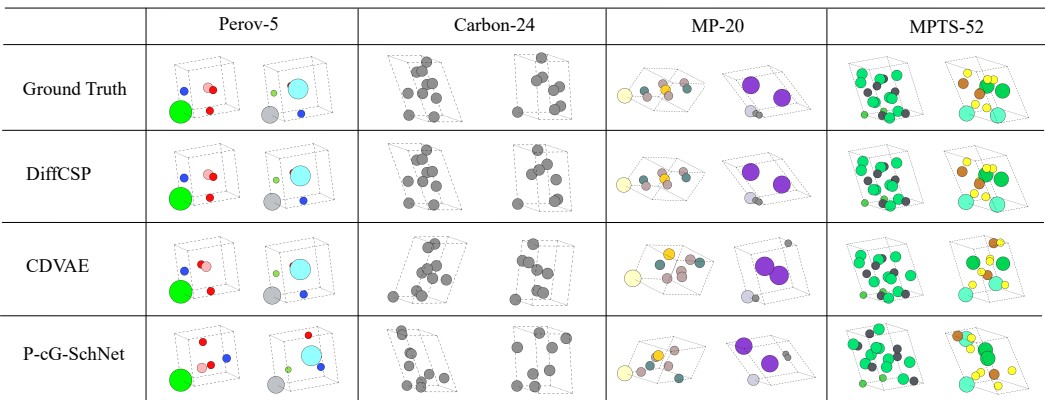

Figure 11: Additional visualizations of the predicted structures from different methods. We translate the same atom to the origin for better visualization and comparison.

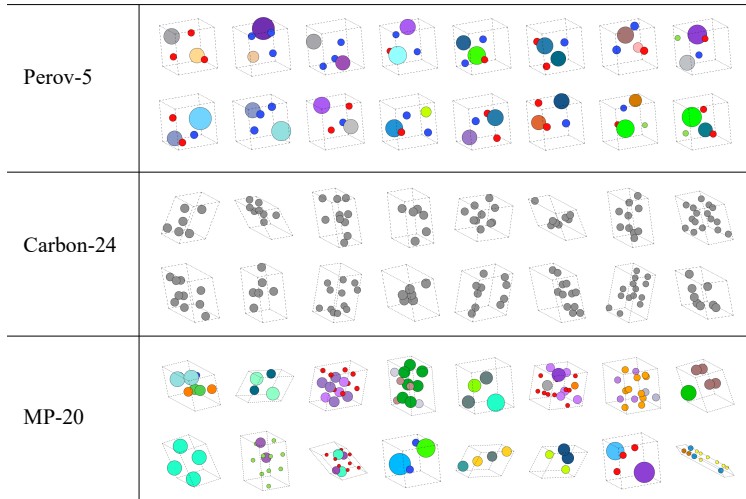

Figure 12: Visualization of the generated structures by DiffCSP.

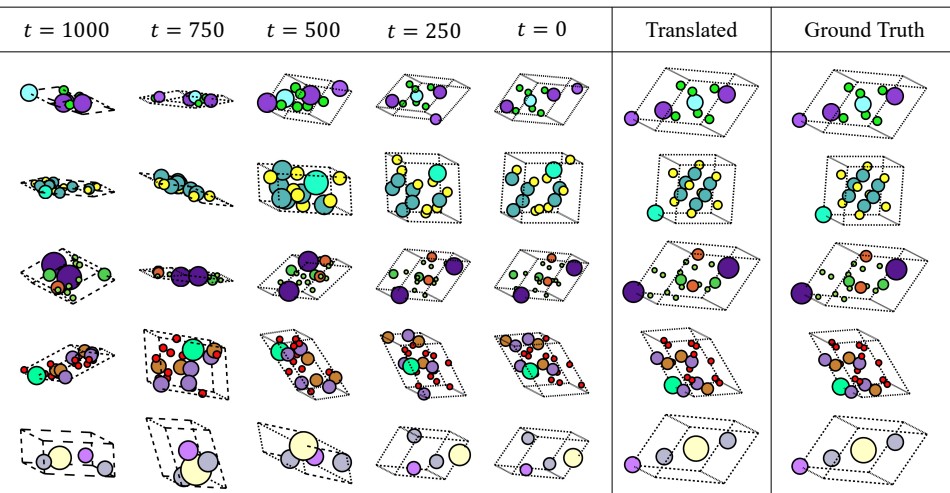

Figure 13: Visualization of the generation process on MP-20. The column "Translated" means translating the same atom in the generated structure $\mathcal{M}_0$ to the origin as the ground truth for better comparison.

