# OpenReview forum: "Crystal Structure Prediction by Joint Equivariant Diffusion"
_NeurIPS.cc/2023/Conference — NeurIPS 2023 poster_

### Official Review · Reviewer_pv4b · 2023-06-28

**Soundness:** 3 good
**Presentation:** 4 excellent
**Contribution:** 2 fair
**Rating:** 5
**Confidence:** 4

**Summary:**

DiffCSP is proposed in this paper for the task of crystal structure prediction task, which generates the stable crystal structures of given compositions. The diffusion process of DIffCSP considers both lattice matrix and fractional coordinates of atoms in the unit cell. Experimental results demonstrate the effectiveness and efficiency of this method.

**Strengths:**

- The proposed method shows good performances compared to previous methods in the task of Crystal Structure Prediction.
- The problem setting is new to ML, and interesting.
- Writing is clear with informative figures, formulas, and proofs.
- Experiments are comprehensive with additional adjustments to change the current framework to the generative model setting and compare with previous generative models. Ablation studies are provided.

**Weaknesses:**

- Limited novelty: the proposed method is straightforward by further considering lattice matrix diffusion; the novelty is somehow limited. The way they diffuse lattice matrix is similar to previous works for molecule conformer generation. And the way they diffuse fractional atom coordinates is similar to CDVAE.
- Other distribution invariant constraints are not considered: (1) The distribution of crystal structures may need to be SO(3) invariant instead of O(3) invariant, given the fact that chiral crystals (similar cases in molecules) are different from each other. (2) Additionally, given the diffusion process proposed, it may not have the ability to deal with the distribution periodic invariance in the following case: given a lattice matrix L = [l_1, l_2, l_3] \in R^{3*3}, the crystal structure remain the same when you change it to L = [l_1, l_2 + l_3, l_3 - l_2], and many other cases when you choose a different set of periodic patterns to describe the same infinite crystal structure.
- Experiments about CSP: adapting CDVAE to conditional generative model and comparing with CSP model is reasonable but not that fair. But they adjust their model to the generative setting and compare with CDVAE, so it may not be a big issue.

**Questions:**

- Maybe a typo: In line 277, it is '20 structures of the lowest energy or' or of?

Suggestions:
- How about rename the Permutation Invariance, O(3) Invariance, etc, to Distribution Permutation Invariance, Distribution O(3) Invariance? The latter ones seem to be more accurate.

**Limitations:**

They listed the limitations in the main paper.

---

> ### Author Rebuttal · Authors · 2023-08-09
>
> We are thankful to the reviewer for the constructive comments. We answer the reviewer’s questions as follows.
>
>
> > **W1: Limited novelty.**
>
> There could be some misunderstandings here. We admit that the way we diffuse lattice matrix is similar to previous works for molecule conformer generation, but we disagree that the way we diffuse fractional atom coordinates is similar to CDVAE. Different from CDVAE that uses a score-matching method on Cartesian coordinates, we utilize fractional coordinates in our diffusion process. As also raised by other reviewers, the usage of fractional coordinates eliminates the need to construct multi-edge graphs (which is used in CDVAE) and naturally guarantees periodic invariance. Given that the domain of fractional coordinates forms a quotient space, we leverage the Score-Matching (SM) based framework along with Wrapped Normal (WN) distribution (Line 182-203).
>
> Moreover, in contrast to CDVAE which focuses solely on coordinate diffusion within a fixed region bounded by the predicted lattice, our model performs joint diffusion on both lattice structures and atom coordinates, and these two processes are coupled and promoted with each other (Figure 2).
>
> Besides, in Section 4.3, we design a novel denoising model to fulfill the joint diffusion process. Firstly, Fourier Transformation of the relative fractional coordinate is adopted to better encode periodicity (Eq.6). Secondly, the output layer is specifically designed to jointly predict the denoising terms of the entire lattice (Eq.9) and each fractional coordinate (Eq.10). Lastly, the proposed denoising model is novel compared to traditional multi-graph representation, and it is proved to be periodic E(3) equivariance (Proposition 3).
>
> Overall, we highlight the novelty of our paper in terms of the three aspects: the diffusion of fractional coordinates via an SM-based framework along with WN distribution, the joint diffusing process, and the novel denoising model.
>
> > **W2.1: Other distribution invariant constraints are not considered.**
>
> Thank you for your valuable suggestions.  We appreciate your insight into the importance of considering SO(3) invariance. In response to this concern, we conduct an ablation study by additionally introducing the chirality into the denoising model. Specifically, we adapt the message-passing block in Eq. (6) into
> $$
> \textbf{m}\_{ij}^{(s)}=\varphi\_m(\textbf{h}\_{i}^{(s-1)}, \textbf{h}\_{j}^{(s-1)}, \textbf{L}^\top\textbf{L}, \text{sign}(|\textbf{L}|), \psi\_{FT}(\textbf{f}\_{j} - \textbf{f}\_{i})),
> $$
> where sign$(|\textbf{L}|)$ is the sign of the determinant of the lattice matrix, which is SO(3)-invariant, but breaks the reflection symmetry and hence is NOT E(3)-invariant. The results of the adapted model on MP-20 are listed in the following table. There is no essential difference between the performance of the SO(3) model and the original E(3) model, which is probably because the chirality is not so crucial in distinguishing different crystal structures for the datasets used in this paper. However, we are willing to add the discussion here to the revised paper.
>
> |                | Match Rate (%)$\uparrow$ |  RMSD$\downarrow$  |
> | ---------------| :------------: | :----: |
> | DiffCSP        |     51.49      | 0.0631 |
> | DiffCSP-SO(3)  |     49.68      | 0.0637 |
>
> > **W2.2: Additionally, given the diffusion process proposed, it may not have the ability to deal with the distribution periodic invariance in the following case...**
>
> Thank you for this nice point. Actually, this invariance is caused by an infinite number of selections for the lattice matrix, to reflect the periodicity of crystal structures. To avoid the exploration of this kind of invariance, we follow the setting from CDVAE and apply the Niggli reduction [46] on the primitive cell as a canonical representation of the lattice vectors, which was mentioned in the footnote of Page 4. We will further clarify this point in the paper.
>
>
> > **W3: Experiments about CSP.**
>
> We understand your concern. Hence, we conducted two kinds of protocols in our experiments, one adapting CDVAE to the settings of the CSP task for fair comparisons with our method (Table 1), the other one extending our method to the setting of CDVAE for ab initio generation (Table 4). Indeed, we discussed the relationship and distinction among various crystal-related tasks in Appendix F.1 and Figure 7.
>
>
>
> We will also fix the raised typos, and further emphasize that the notions of the Permutation Invariance, O(3) Invariance, etc., are defined in terms of distribution.

---

### Official Review · Reviewer_pLZt · 2023-07-01

**Soundness:** 3 good
**Presentation:** 3 good
**Contribution:** 4 excellent
**Rating:** 8
**Confidence:** 4

**Summary:**

DiffCSP is a novel diffusion model for predicting crystal structures conditioned on the crystal's atom composition; it can also be extended to an initio crystal generation (sampling both atom composition and positions). Crystal modeling is challenging due to its periodic symmetry. DiffCSP proposes to solve periodicity constraints by diffusing on fractional coordinates and jointly generating the lattice and atom coordinates. Fractional coordinate diffusion is achieved with torsional diffusion that constraints each intermediate state to stay within a bounding box, i.e. a fixed unit cell. When compared with existing methods, DiffCSP is state-of-the-art across DFT and ML-based methods in predicting the crystal structures on three datasets -- Perov-5, Carbon-24, and MP-20 -- while requiring less computation. It is also state-of-the-art on ab initio generation for the same datasets.

**Strengths:**

- Using fractional coordinates to handle periodicity is supported by several prior studies on crystal structures. It is encouraging to see DiffCSP explore this representation in diffusion models. The authors do a in-depth comparison with CDVAE [1] which alternatively explored a multi-graph construction for handling periodicity.

- DiffCSP studies joint diffusion over all three components of crystal structures: atom types, lattice matrix, and coordinates. The comparisons and benchmarks will be useful to the broader community.

- The denoising model, a modified EGNN, to achieve the periodic E(3) symmetry has architectural choices that are well motivated. Each component is ablated to demonstrate their individual importances.

- DiffCSP achieves state-of-the-art in crystal structure prediction and ab initio crystal generation.


[1] https://arxiv.org/abs/2110.06197
[2] https://arxiv.org/pdf/2102.09844.pdf

**Weaknesses:**

- The authors claim DiffCSP is the first generative model for crystal structure prediction (CSP). I am curious to know why a generative viewpoint of CSP is beneficial. Discussion of why diffusion is important for this task would help, i.e. a discussion about uncertainty in the problem and how it manifests.

- The benefit of diffusing over the lattice could use more explanation. The ablation seems to show there is marginal benefit of diffusing the lattice. It is also not clear to me if gaussian diffusion is the right choice for the lattice vectors which can also be parameterized with 6 unique, rotation invariant parameters using the Niggli algorithm [1].

- DiffCSP is E(3) and periodicity equivaraint however CDVAE is SE(3) and periodicity equivariant. Perhaps I missed it but I do not see discussion on the subtley between E(3) and SE(3). It would be important to note this difference and include it in the discussion.

- A major consideration when performing joint diffusion is the noise schedules used for each diffusion process. A discussion of whether this was important practically would be helpful in the main text.



[1] https://pubmed.ncbi.nlm.nih.gov/14691322/

**Questions:**

Concrete questions pertaining to my weaknesses:

- If there's a single right answer of the structure for a given atom composition then is it right to generate many possible structures? The author evaluate DiffCSP and other generative models by matching the samples to the ground truth structure to pick the best sample. But how would one select the best sample if the ground truth was not available?

- What is the rationale behind gaussian diffusion for the lattice parameters? Are magnitudes of the lattice bounded? If it is unbounded then it seems strange to use gaussian diffusion.

- Why is DiffCSP chosen to be E(3) equivariant but not SE(3) equivariant like in CDVAE? Is this a limitation?

- Was the noising schedule of the atom types, lattice, and coordinates important? Did it make a practical difference?

Other questions:

- Could Table 1 include mean and standard deviation for RMSE? If it doesn't fit then in the appendix would be fine. It would be helpful to know the spread of the metrics when multiple samples are used.

- How diverse are the crystals when multiple samples are drawn?



Typoes/suggestions:

- Line 33. Reference 11 does protein structure generation, not prediction.

- It would be helpful to include mean and standard deviation for RMSE in Table 1. If it doesn't fit then in the appendix would be fine. It would be helpful to know the spread of the metrics when multiple samples are used.

- It is odd to frame diffusion on the fractional coordinates as a DDPM when it is really a VE-SDE following the SDE formulation. I believe it will cause less confusion if some clarification is given to the original source of the torsional diffusion formulation as Riemannian score-matching [1].

[1] https://arxiv.org/abs/2202.02763


**Limitations:**

- A potential limitation is SE(3) vs. E(3) equivariance. The authors do not discuss this. I have mentioned this already in the questions.

- The authors do not discuss choice of diffusion hyperarameters which I believe would have lots of practical use for researchers who would build on top of this work.

---

> ### Author Rebuttal · Authors · 2023-08-09
>
> We are thankful to the reviewer for the detailed comments. We answer the reviewer’s questions as follows.
>
> For weaknesses:
>
> > **W1: Why a generative viewpoint of CSP is beneficial.**
>
> Thank you for your question. The generative viewpoint in CSP holds substantial benefits, particularly in addressing inherent uncertainties within the problem and capturing the underlying distribution of possible crystal structures. One of the fundamental challenges in CSP is the non-uniqueness of crystal structures, even when the composition is given and fixed. A classic example is carbon, which can form various allotropes, each with distinct structures (diamond, graphite, etc.). DiffCSP acknowledges this inherent diversity and leverages diffusion processes to explore the rich space of possible structures.
>
> > **W2: The benefit of diffusing over the lattice could use more explanation.**
>
> Thank you for reminding. To address your concern, we provide a variant model named DiffCSP-L, which utilizes a Transformer-based predictor to directly predict the 6 invariant parameters from a given composition. Table R1 (General Response) displays the results on MP-20, highlighting the diffusion model's superior performance over the predictor-based model.  We conjecture that the distribution of crystals depends on both the lattice and the atom coordinates within each unit cell. By diffusing both, the generative model is able to simultaneously adjust their values, leading to more accurate generation. By contrast, the predictor-based variant conducts a two-stage generation paradigm, and the lattice is no longer adjustable during the generation of the atom coordinates. We will add the above explanation to the revised version.
>
> > **W3: SE(3) v.s. E(3) invariance.**
>
> Great suggestion! We attempt to introduce the chirality into the denoising model by adapting the message-passing block in Eq. (6) into
> $$
> \textbf{m}\_{ij}^{(s)}=\varphi\_m(\textbf{h}\_{i}^{(s-1)}, \textbf{h}\_{j}^{(s-1)}, \textbf{L}^\top\textbf{L}, \text{sign}(|\textbf{L}|), \psi_{FT}(\textbf{f}\_{j} - \textbf{f}\_{i})),
> $$
> where sign$(|\textbf{L}|)$ is the sign of the determinant of the lattice matrix, which is SO(3)-invariant, but breaks the reflection symmetry and hence is NOT E(3)-invariant. The results on MP-20 are listed in Table R1 (General Response). There is no essential difference between the performance of the SO(3) model and the original E(3) model, which is probably because the chirality is not so crucial in distinguishing different crystal structures for the datasets used in this paper.
>
> > **W4: The noise schedules used for each diffusion process.**
>
> Thanks for your suggestion. We explore the noise schedulers from 3 perspectives and list the results in Table R2 and R3 (General Response). 1) For lattices, we originally use the cosine scheduler with $s=0.008$, and we change it into the linear and sigmoid schedulers with $\beta_1=0.0001, \beta_T=0.02$. We find that the linear scheduler yields comparable results, while the sigmoid scheduler hinders the performance. 2) For coordinates, we use the exponential scheduler with $\sigma_{min}=0.005, \sigma_{max}=0.5$, and we change $\sigma_{max}$ into 0.1 and 1.0. Results show that the small-$\sigma$ variant performs obviously worse, as only sufficiently large $\sigma$ could approximate the prior uniform distribution. We visualize the PDF curves in Figure R3 for better understanding. 3) For types, we conduct similar experiments as lattices, and the results indicate that the original cosine scheduler performs better. In conclusion, we suggest applying the proposed noise schedulers, and we will provide the above details in the appendix.
>
> For questions (Note that we have merged some repeated questions due to space limitation):
>
> > **Q1: How would one select the best sample if the ground truth was not available?**
>
> Nice question! If the ground truth is unavailable, ranking the generated samples is indeed an important problem. To this end, we here design 3 kinds of confidence models to score each sample for ranking, named Energy Predictor(EP), Match Discriminator(MD), and Contrastive Scorer(CS). See General Response for more details.
>
> We select the Top-1 result among the 20 candidates by each confidence model on MP-20 in Table R4 (General Response). It shows that MD and CS perform relatively better.  We agree that structure ranking is crucial and are willing to add the above discussion to the revised paper.
>
> > **Q2: What is the rationale behind gaussian diffusion for the lattice parameters?**
>
> Thank you for this insightful question. The magnitudes of the lattice lengths are within [3.29, 5.71], [2.40, 16.67], [2.25, 46.75], and [0.98, 130.46] on Perov-5, Carbon-24, MP-20, and MPTS-52, respectively. In this sense, the magnitudes are practically bounded, and we apply gaussion diffusion for the lattice vectors since their values are real and continue.
>
> > **Q3: Could Table 1 include mean and standard deviation for RMSE?**
>
> Yes, we did provide the error bars in Table 11 of Appendix H. We will further clarify this point in the main paper.
>
> > **Q4: How diverse are the crystals when multiple samples are drawn?**
>
> Good question! We further evaluate the diversity by yielding the CrystalNN fingerprint of each generated structure, calculating the L2-distances among all pairs in the 20 samples of each composition, collating the mean and max value of the distances, and finally averaging the results from all testing candidates. The results are listed in Table R5 (General Response). Clear diversity is observed for both CDVAE and our DiffCSP.
>
> For typoes & discussions:
>
> Thanks for your suggestions. We are glad to modify the typoes and provide more discussions on Riemann score-matching. The error bars were provided in Table 11 of Appendix H.
>
> For limitations:
>
> Thanks for your kind suggestions. We have explored the influence of SO(3) invariance (W3) and noise schedulers (W4). We will add these discussions to the revised paper.

---

> > ### Comment · Reviewer_pLZt · 2023-08-10
> > **Response**
> >
> > Thank you for the very detailed response. I have read the global rebuttal and enjoyed all the additional analysis. I have raised my score as a result. I believe this problem formulation will be a useful and interesting contribution to NeurIPS.

---

> > > ### Author Response · Authors · 2023-08-12
> > > **Thank you**
> > >
> > > Dear Reviewer pLZt,
> > >
> > > Thank you very much! Your valuable comments and suggestions have helped improve the paper. Thanks again!
> > >
> > > Best,
> > >
> > > Authors

---

### Official Review · Reviewer_eNoa · 2023-07-03

**Soundness:** 3 good
**Presentation:** 3 good
**Contribution:** 3 good
**Rating:** 7
**Confidence:** 4

**Summary:**

This paper addresses the crystal structure prediction problem, one of the important problems in this field.
The authors propose a diffusion model generating both lattices and fractional coordinates given chemical composition.
The proposed denoising model satisfies O(3) equivariance and periodic translation invariance.
The experimental results show that the proposed method outperforms the conventional methods in generating existing crystals.

**Strengths:**

* The paper is well-organized and well-written.
* The representation of crystals is reasonable.
* The empirical result is strong, and the baseline methods are appropriate for the CSP task.
* The results compared with USPEX, a strong GA-based method,  attract the CSP community's interest.
* Ablation studies support the design choice of the proposed method.

**Weaknesses:**

The study does not report any new crystal structures. It will increase the impact of this paper if the proposed method can find a new crystal that achieves lower energies via DFT than that of the known crystals of a given multi-component system.

A minor comment

"$\psi_L$ in Eq. (10)" in the main text of section 4.3 seems to be a typo in Eq. (9).

**Questions:**

Is it reasonable to compare the inference time of the proposed method with the computation time of USPEX? Should the training time be included in the CSP time? How long does the training take?

**Limitations:**

Yes, the limitation is discussed in Section 6.

---

> ### Author Rebuttal · Authors · 2023-08-09
>
> We are thankful to the reviewer for the constructive comments. We answer the reviewer’s questions as follows.
>
> > **Q1: The study does not report any new crystal structures. It will increase the impact of this paper if the proposed method can find a new crystal that achieves lower energies via DFT than that of the known crystals of a given multi-component system.**
>
> Nice suggestion! In fact, we have equipped our method with a guidance model to generate structures with lower energies. Please refer to Appendix F.3 for more details. Here according to the reviewer's suggestion, we fix the composition of the initial crystal and only optimize its structures for lower energy. Results are visualized in Figure R2 (General Response), where the energies of the structures before and after optimization by VASP are shown. It reads that our method does search novel structures with lower energies compared to existing ones.
>
>
> > **Q2: Is it reasonable to compare the inference time of the proposed method with the computation time of USPEX? Should the training time be included in the CSP time? How long does the training take?**
>
> Thanks for pointing it out. In our analysis of inference time, we aim to focus on the efficiency of generating samples once the model has been trained and is ready for deployment. This specific aspect is crucial when considering the practical application of a method in real-world scenarios, where users are primarily concerned with obtaining results in a timely manner. While the training time is indeed an essential consideration in understanding the overall computational cost of a method, it represents a one-time investment that occurs during the model development phase. Once the model is trained, its inference time becomes a recurring factor every time predictions are needed.
>
> Moreover, the training time for our method is approximately 5.2 hours on MP-20, and we do acknowledge that this is an important consideration for users when implementing our model. Even taking the training time into consideration, our method still showcases more efficiency against USPEX (the total time is 5.2h + 10s vs 12.5h).

---

> > ### Comment · Reviewer_eNoa · 2023-08-14
> > **Thanks for the comments**
> >
> > Thank you for your reply.
> >
> > > Nice suggestion! In fact, we have equipped our method with a guidance model to generate structures with lower energies. Please refer to Appendix F.3 for more details. Here according to the reviewer's suggestion, we fix the composition of the initial crystal and only optimize its structures for lower energy. Results are visualized in Figure R2 (General Response), where the energies of the structures before and after optimization by VASP are shown. It reads that our method does search novel structures with lower energies compared to existing ones.
> >
> > It sounds promising, but the impact of this paper could be enhanced more by listing novel crystal structures that possess lower energy than those currently registered in crystal databases, such as the Crystallography Open Database or Material Project.
> >
> > > Moreover, the training time for our method is approximately 5.2 hours on MP-20, and we do acknowledge that this is an important consideration for users when implementing our model. Even taking the training time into consideration, our method still showcases more efficiency against USPEX (the total time is 5.2h + 10s vs 12.5h).
> >
> > The total time, i.e., 5.2h + 10s, is still impressive, and I recommend including the training time in the main text since many readers would like to know it.

---

> > > ### Author Response · Authors · 2023-08-15
> > >
> > > Thank you very much for your feedback.
> > >
> > > > It sounds promising, but the impact of this paper could be enhanced more by listing novel crystal structures that possess lower energy than those currently registered in crystal databases, such as the Crystallography Open Database or Material Project.
> > >
> > > We really agree with your suggestion of enhancing our paper by listing novel crystal structures with lower energy compared with the Crystallography Open Database [1] or Material Project [2], which is actually our ultimate goal of developing generative models for the CSP task. Given that our current framework can only optimize the structure of the input crystal composition one by one, it would be more challenging to discover a totally novel and more stable structure in constrast to all structures in the Crystallography Open Database and Material Project that contain a vast number of structures (504,942 and 144,595 entries respectively). Optimizing each individual structure with DFT verification can be a time-consuming process, and it may also require larger datasets to train a more generalized diffusion model. While we do understand and appreciate the kind suggestion by the reviewer, we are more willing to leave it for future studies.
> > >
> > > [1] http://www.crystallography.net/cod/
> > >
> > > [2] https://legacy.materialsproject.org/
> > >
> > > > The total time, i.e., 5.2h + 10s, is still impressive, and I recommend including the training time in the main text since many readers would like to know it.
> > >
> > > Thank you once again. We will certainly integrate the training time into the revised version of the main paper.

---

> > > > ### Comment · Reviewer_eNoa · 2023-08-15
> > > > **Thank you**
> > > >
> > > > Thank you, and I am looking forward to seeing follow-on research in the future.

---

### Official Review · Reviewer_1HBn · 2023-07-08

**Soundness:** 3 good
**Presentation:** 3 good
**Contribution:** 3 good
**Rating:** 5
**Confidence:** 5

**Summary:**

The paper studies crystal structure prediction, an important topic in the field of AI for scientific discovery. Diffusion models are designed considering necessary symmetries for crystal study, with diffusing lattice vectors and fractional coordinates for generating lattices and atoms, respectively. Experiments show the effectiveness of the proposed method.

**Strengths:**

1. The paper presents novel and shining ideas, e.g., considers fractional coordinates rather than Cartesian coordinates. This eliminates the need of constructing multi-edge graphs and naturally guarantees periodic invariance. The theory for processing fractional coordinates in the diffusion model looks rigid and solid.

2. The joint learning fashion for lattices and coordinates should be the right way to go in the field.

3. Experiments are solid and performance is impressive.

**Weaknesses:**

1. One major concern is-- the technical contribution is not clear or strong enough. Diffusion models for 3D scientific data ate not new. The authors may need to clearly state the significant and unique design of the proposed diffusion model to distinguish it from other diffusion models.

2. Another concern is-- necessary symmetries are not completely considered. For example, similar to CDVAE, translation invariance (not periodic translation invariance) may not be considered.

3. A key foundation is-- the consideration of fractional coordinates can achieve periodic invariance, which is roughly correct. However, if the three lattice vectors are not defined in a deterministic way, this may not be true.

4. There are some issues in the organization. For example, much content in 4.1 overlaps with ref [36], and this content should be put in background/preliminary and credits should be given properly.

Overall, this is a neat work. I would raise my scores if my concerns are fully addressed.

**Questions:**

See Weaknesses. An additional one is-- how's the efficiency compared with CDVAE in empirical studies?

---

> ### Author Rebuttal · Authors · 2023-08-09
>
> We are thankful to the reviewer for the constructive comments. We answer the reviewer’s questions as follows.
>
> > **Q1: The technical contribution is not clear or strong enough. Diffusion models for 3D scientific data are not new. The authors may need to clearly state the significant and unique design of the proposed diffusion model to distinguish it from other diffusion models.**
>
> Sorry for the insufficient clarity regarding our technical contribution. We sincerely thank the reviewer and will further clarify our contributions with the statements below:
>
> Even diffusion models for GENERAL 3D molecules [41] are not new, this paper focuses on crystals which exihibit an extra form of symmetry, namely, periodicity. As a consequence, our diffusion models should not only generate the 3D coordinates but also the lattice matrix. This leads to the unique design of our model:
> - **Joint Diffusion Framework**: Our model performs joint diffusion on both lattice structures and atom coordinates, and these two processes are coupled and promoted with each other (Figure 2). Conversely, prior crystal diffusion frameworks such as CDVAE[36] focus solely on coordinate diffusion within a fixed region bounded by the predicted lattice.
> - **Periodicity-Aware Design**: While previous models for crystals[36] employ Cartesian coordinates, we depart from this convention by utilizing fractional coordinates, which naturally fit the periodicity consideration. Given that the domain of fractional coordinates forms a quotient space, we leverage the Score-Matching (SM) based framework along with Wrapped Normal distribution (Line 182-203), other than the commonly-used DDPM paradigm.
> - **Novel Denoising Model**: In Section 4.3, we design a novel denoising model to fulfill the above diffusion process. Firstly, Fourier Transformation of the relative fractional coordinate is adopted to better encode periodicity (Eq.6). Secondly, the output layer is specifically designed to jointly predict the denoising terms of the entire lattice (Eq.9) and each fractional coordinate (Eq.10). Lastly, the proposed denoising model is novel compared to traditional multi-graph representation, and it is proved to be periodic E(3) equivariance (Proposition 3).
>
> The above statements will be added to the revised paper.
>
> > **Q2: Necessary symmetries are not completely considered. For example, similar to CDVAE, translation invariance (not periodic translation invariance) may not be considered.**
>
> Thank you for the comment. Actually, for crystals, our definition of periodic translation invariance already implies translation invariance. To see this, we provide an example in Figure R1 (General Response). From a global view, when we translate all atom coordinates from left to right, the crystal structure remains unchanged, which indicates translation invariance. At the same time, from the view of a unit cell, the atom translated across the right boundary will be brought back to the left side owing to periodicity ($L$ is fixed), which indicates periodic invariance. Therefore, for convenience, we define the joint effect of translation invariance and periodicity as periodic translation invariance. We will further clarify this point in the paper.
>
> > **Q3: The consideration of fractional coordinates can achieve periodic invariance, which is roughly correct. However, if the three lattice vectors are not defined in a deterministic way, this may not be true.**
>
> We are unsure if we understand the reviewer's question (please point it out if not). Our point is that the definition of periodic invariance is independent of whether the lattice vectors are deterministic or not. When defining periodic invariance, we mean given a lattice matrix, the crystal distribution remains the same under any global translation of atom coordinates. This can be checked in the equation in Definition 3, where both $L$ and $F$ are fixed and only the translation $t$ varies.
>
>
> > **Q4: There are some issues in the organization. For example, much content in 4.1 overlaps with ref [36], and this content should be put in background/preliminary and credits should be given properly.**
>
> Thanks for pointing it out. We are willing to move Section 4.1 to "Background/Preliminary", and will further emphasize the connections and differences with [36]. Indeed we have discussed the difference between our definitions and those presented in [36] in Appendix A.4, which will be modified into the main paper.
>
> > **Q5: How's the efficiency compared with CDVAE in empirical studies?**
>
> Thank you for the comment. We have indeed compared the computational cost of each generative method in Appendix G and Table 9. We provide the GPU hours (GeForce RTX 3090) for different generative methods to predict 20 candidates on the 4 datasets. Our method is more efficient mainly because it requires fewer diffusion steps for generation. Particularly, to achieve the accuracies reported in Table 1, CDVAE requires 5,000 steps for each generation, whereas our approach only requires 1,000 steps. We bring the results here for convenience.
>
> |             | Perov-5 | Carbon-24 | MP-20 | MPTS-52 |
> |-------------|---------|-----------|-------|---------|
> | CDVAE       | 21.9    | 9.8       | 95.2  | 178.0   |
> | DiffCSP     | 1.5     | 3.2       | 18.4  | 34.0    |

---

> > ### Comment · Reviewer_1HBn · 2023-08-13
> >
> > Given the clarifications the authors provided as well as the promise that certain content will be added in the revision, I raise my rating to 5. Considering the work is not significant enough in the context of existing studies, 5 is the best score I can give.

---

> > > ### Author Response · Authors · 2023-08-15
> > > **Thank you**
> > >
> > > Thanks for raising the score. We will highlight the contributions in the revised paper.

---

### Author Rebuttal · Authors · 2023-08-09

We sincerely thank all reviewers and ACs for their time and efforts on reviewing the paper. We are glad that the reviewers recognized the contributions of our paper, which we briefly summarize as follows.

- **Novelty.** "The paper presents novel and shining ideas."(1HBn)"The problem setting is new to ML, and interesting."(pv4b)
- **Presentation.** "The paper is well-organized and well-written."(eNoa)"Writing is clear with informative figures, formulas, and proofs."(pv4b)
- **Experiments.** "Experiments are solid and performance is impressive."(1HBn)"The empirical result is strong, and the baseline methods are appropriate for the CSP task."(eNoa)"The comparisons and benchmarks will be useful to the broader community."(pLZt)"Experiments are comprehensive with additional adjustments to change the current framework to the generative model setting and compare with previous generative models."(pv4b)

We also appreciate the reviewers for their insightful comments, and provide additional visualizations and experment results in the supplementary PDF file for more details. We summarize the extra contents as follows.

- **Figure R1 (to Reviewer 1HBn)** illustrates the connection between the translation invariance in previous works and the periodic translation invariance defined in our paper. The periodic translation invariance already considers the translation invariance.
- **Figure R2 (to Reviewer eNoa)** visualizes 8 pairs of original and optimized structures, indicating that DiffCSP is capable to obtain structures with lower energies from existing ones.
- **Figure R3 (to Reviewer pLZt)** compares the PDF curves of wrapped normal distributions with period 1 and diffenent variances. It can be find that $\sigma=0.5$ is practically large enough to approximate the prior uniform distribution.
- **Table R1 (to Reviewer pLZt and pv4b)** conducts two extra ablation studies. The first explores the necessity of lattice diffusion against a predictor-based counterpart, and the second compares the performances of SO(3) and O(3) equivariant backbones.
- **Table R2 and R3 (to Reviewer pLZt)** evaluate the influence of noise schedulers of the diffusion processes on lattices, coordinates and atom types.
- **Table R4 (to Reviewer pLZt)** compares different confidence models for ranking candidate structures. Specifically, we design 3 kinds of confidence models:

1) **Energy Predictor(EP).** We directly train a predictor of the energy labels, and apply the opposite of the predicted energy as the confidence score.
2) **Match Discriminator(MD).** Inspired by Diffdock[1], we first generate 5 samples for each composition in the train/validation set via DiffCSP, and calculate their RMSDs with the ground truth. Finally, we train a binary classifier to predict whether the sample matches the ground truth with RMSD less than a threshold $d$. The predicted probability is used as the score.
3) **Contrastive Scorer(CS).** Inspired by CLIP[2], we train a contrastive model between a 1D and a 3D model to align the corresponding compositions and structures, and the inner product of the 1D and 3D model is used as the score.

- **Table R5 (to Reviewer pLZt)** compares the diversity of generated structures of CDVAE and DiffCSP.

[1] Corso, Gabriele, et al. "DiffDock: Diffusion Steps, Twists, and Turns for Molecular Docking." International Conference on Learning Representations (ICLR 2023). 2023.
[2] Radford, Alec, et al. "Learning transferable visual models from natural language supervision." International conference on machine learning. PMLR, 2021. similar tothan other schedulers

---

### Decision · Program_Chairs · 2023-09-21

**Decision:**

Accept (poster)

**Comment:**

This paper studies crystal structure prediction which is an important application of AI for scientific discovery. The newly proposed diffusion model is equipped with nice ideas (fractional coordinates and consideration symmetries) and the experimental evaluation is concrete. This paper is likely to influence more researchers to study the problem of neural crystal structure prediction.

The reviewers are all positive about the paper, appreciating the importance of the problem, empirical setup, and novelty of the fractional coordinates. I vote for acceptance of this work.